

# 1 Multi-scale water balance analysis of a thawing boreal peatland
# 2 complex near the southern permafrost limit in western Canada

Alexandre Lhosmot[1*], Gabriel Hould Gosselin[1,2*], Manuel Helbig[1,3], Julien Fouché[1,4], Youngryel Ryu[5]
Matteo Detto[6], Ryan Connon[7], William Quinton[8], Tim Moore[9] and Oliver Sonnentag[1,10]
[1]Département de géographie, Université de Montréal, Montréal, QC, Canada
[2]Department of Geography and Environmental Sciences, Northumbria University, Newcastle upon Tyne, UK
[3]Department of Physics & Atmospheric Science, Dalhousie University, Halifax, NS, Canada
[4]LISAH, Université de Montpellier, INRAE, IRD, Institut Agro, AgroParisTech, Montpellier, France
[5]Department of Landscape Architecture and Rural Systems Engineering, Seoul National University, Seoul, South
Korea
[6]Department of Ecology and Evolutionary Biology, Princeton University, Princeton, NJ, USA
[7]Environment and Climate Change, Government of the Northwest Territories, Yellowknife, NT, Canada
[8]Cold Regions Research Centre, Wilfrid Laurier University, Waterloo, ON, Canada
[9]Department of Geography, McGill University, Montréal, QC, Canada
[10]Department of Geography and Environmental Studies, Wilfrid Laurier University, Waterloo, ON, Canada
*These authors share the co-first authorship.
*Correspondence to:* Alexandre Lhosmot (alexandrelhosmot@gmail.com) and Oliver Sonnentag
(oliver.sonnentag@umontreal.ca)
**Abstract.** Permafrost thaw profoundly changes landscapes in the Arctic-boreal region, affecting ecosystem
composition, structure, function and services and their hydrological controls. The water balance provides insights
into water movement and distribution within a specific area and thus helps understand how different components
of the hydrological cycle interact with each other. However, the water balance of small- ($<10^1$ km²) and meso-scale
basins ($10^1$-$10^3$ km²) in thawing landscapes remains poorly understood. Here, we conducted an observational study
in three small-scale basins (0.1-0.3 km²) of a thawing boreal peatland complex. The three small-scale basins were
situated in the Scotty Creek basin headwater portion, a meso-scale low-relief basin (drainage area estimates from
130 to 202 km²) near the southern permafrost limit in the Taiga Plains ecozone in western Canada. By measuring
water losses (discharge, evapotranspiration [ET]), inputs (rainfall [R], snow water equivalent [SWE]) and storage
change (ΔS), and calculating runoff (Q), we (1) aimed at quantifying growing season (May-September, 2014-2016)
headwater small-scale basins water balances, i.e., sub-basins. After (2) comparing monthly sub-basin- and
corresponding basin water losses through ET and Q, we aimed at (3) assessing the long-term (1996-2022) annual
basin water balance using publicly available observations of discharge (and thus calculated Q), R and SWE in
combination with simulated ET. (1) Growing season water balance residuals (RES) for the sub-basins ranged from





-81 to +122 mm. The monthly growing season water balance for the sub-basin for which all the water balance
components throughout the three-year study period were recorded exhibited large positive RES for May (+117 to
+176 mm) since it included late-winter SWE routinely estimated in late March right before snowmelt. In contrast,
lower monthly RES were obtained from June to September (-41 to 0 mm). For two sub-basins, we provide two
different drainage area estimates highlighting the challenge of automated terrain analysis using digital elevation
models in low-relief landscapes. Drainage areas were similar for one sub-basin but exhibited a fivefold difference
for the other. This discrepancy was attributed to the high degree of landscape heterogeneity and resulting
hydrological connectivity with implications for Q calculations and RES. (2) The spring freshet contributed 41 to
100 % (sub-basins) and 50 to 79 % (basin) of the April-September Q. Spring freshet peaks were comparable, except
for the driest year (2014), when basin Q was more than ten times lower than in the sub-basins. At both scales ET
was the dominating water loss, more than twice Q. (3) Over the long-term (1996-2022), the increase of basin runoff
ratio (ratio of runoff to precipitation) from 1996 to 2012 (0.1 to 0.5) has been attributed to the increasing
connectivity of wetlands to the drainage network caused by permafrost thaw. However, the smaller average and
more variable runoff ratio from 2013 to 2022 may be due to wetland drying and/or changes in precipitation patterns.
Long-term hydrological monitoring is crucial to identify and understand potential threshold effects (e.g.,
hydrological connectivity) and ecohydrological feedbacks affecting local (e.g., subsistence activities), regional
(e.g., weather) and global ecosystem services (e.g., carbon storage) provided by thawing boreal peatland
complexes.

**Key words: headwater sub-basin, water balance, landscape, runoff, automated terrain analysis, digital**
**elevation model, evapotranspiration, eddy covariance, permafrost, hydrological connectivity**
**1    Introduction**

A large portion of the Arctic-boreal region is characterized by permafrost (perennially frozen ground).

Understanding interactions between permafrost thaw induced landscape changes and hydrological processes is
critical for predicting changes in ecosystem composition, structure, function and services in response to climate
change (Walvoord and Kurylyk, 2016). Permafrost coverage varies widely across the Arctic-boreal region and
increases with latitude and/or altitude (Gruber, 2012). The maximum thickness of the seasonally thawed and
hydrologically active layer above the permafrost generally decreases from the southern permafrost limit northwards
(Ran et al., 2022). Active layer thickness, partly controlled by local climate, ecosystem characteristics and ground





properties (e.g., porosity, water content) ranges approximately from more than one meter (~60 °N) to less than 0.5
m (~70 °N) across Canada (Ran et al., 2022). Higher water content, by simultaneously increasing the latent heat of
fusion during thaw and enhancing thermal conductivity, has an opposite effect on the active layer thickness. The
latent heat of fusion exerts a stronger control on active layer thickness, leading to a thinner active layer (Clayton et
al., 2021). For example, in saturated peat deposits with a porosity of about 80 % at 61°N latitude, the active layer
thickness did not exceed 0.8 m (Connon et al., 2018).

In recent decades, the Arctic-boreal region has experienced a rapid increase in air temperature, up to four

times greater than on a global scale (Rantanen et al., 2022). This atmospheric warming has led to accelerated
permafrost thaw (Biskaborn et al., 2019; Smith et al., 2022). Additional factors, including natural (e.g., wildfires)
and anthropogenic disturbances (e.g., extractive activities; Foster et al., 2022; Klotz et al., 2023), were shown to
increase ground heat flux thus accelerating permafrost warming and thaw (Gibson et al., 2018; Li et al., 2021).
Recent scientific advances have provided insights into the multifaceted and interdependent ecological,
hydrological, atmospheric, and biogeochemical consequences of permafrost thaw (e.g., Burd et al., 2018; Carpino
et al., 2021; Gordon et al., 2016; Quinton et al., 2019; St. Jacques and Sauchyn, 2009; Torre Jorgenson et al., 2013).
In addition, permafrost thaw presents a substantial socio-environmental challenge in the 21st century (Pi et al.,
2021; King et al., 2018). For example, accelerated permafrost thaw threatens local communities, infrastructure, and
Indigenous livelihoods and cultural practices across the northern circumpolar permafrost region (Gibson et al.,
2021; Langer et al., 2023).

From hydrological and biogeochemical perspectives, permafrost thaw has the potential to cause changes

in land cover and hydrological connectivity, and thus in how water and matter moves across and through the
changing landscapes of the Arctic-boreal region (Box et al., 2019; Walvoord and Kurylyk, 2016; Wright et al.,
2022). For example, thaw induced changes in land cover and hydrological connectivity potentially affect
composition and export of both particulate and dissolved organic carbon (Burd et al., 2018; Vonk et al., 2015),
mercury methylation (Gordon et al., 2016), or sulphide oxidation and weathering (Kemeny et al., 2023). Additional
complexity is added through changes in precipitation regimes, projected to shift from snow- to rainfall-dominated
at least in parts of the Arctic-boreal region (He and Pomeroy, 2023; Thackeray et al., 2022). A better thawing
landscapes hydrological understanding in the Arctic-boreal region is crucial to predict the permafrost-carbon
feedback strength at global scale (Ramage et al., 2024; Schuur et al., 2022; Treat et al., 2024).

In the Taiga Plains ecozone of western Canada, which covers approximately 550,000 km², permafrost

coverage varies between isolated (<10 % coverage), sporadic (10 - <50 %) and discontinuous (50 - <90 %)
(Ecosystem Classification Group, 2007; Wright et al., 2022). There, a large portion of the low-relief landscape



comprises boreal peatland complexes including black spruce (*Picea mariana*)-dominated permafrost peat plateaus
and permafrost-free, treeless wetlands resulting from surface subsidence due to ground ice melt (i.e., thermokarst;
Wright et al., 2022). Thermokarst wetlands form depressions and receive water from surrounding permafrost peat
plateaus and some are connected to the drainage network and basin outlet through channel fens. Since the 1970s,
the faster thaw rate of ground ice-rich permafrost has resulted in the expansion of wetlands at the expense of forests
especially near the southern permafrost limit in the southern Taiga Plains (Chasmer and Hopkinson, 2017; Wright
et al., 2022). There, permafrost thaw was found as an equal driver of boreal forest loss as wildfire (Helbig et al.,
2016a). For example, from 1970 to 2010, forests transformed into wetlands at rates ranging from 6.9 % to 11.6 %
across ten sites, each covering 10 km² and spanning from 59.97 °N to 61.3 °N (Carpino et al., 2018). This prominent
thaw induced land cover change has increased hydrological connectivity across the boreal peatland complexes
(Connon et al., 2014; 2015; Quinton et al., 2019) and modified the water balances of small- and meso-scale basins,
$<10^1$ km² and $10^1$-$10^3$ km², respectively (Carey et al., 2010; Uhlenbrook et al., 2004).
Understanding the water balances of small- and meso-scale basins is essential for assessing the
hydrological responses at broader, regional scales (Evenson et al., 2018; Zhang et al., 2018). In the boreal context,
studies have focused specifically on evapotranspiration (ET; Helbig et al., 2016b; Isabelle et al., 2018; Warren et
al., 2018) or runoff (Q; Connon et al., 2014; Mack et al., 2021; St. Jacques and Sauchyn, 2009). In some studies,
Q or water storage changes (ΔS) were obtained as water balance residual (RES), or ET was estimated with a
hydrochemical method or empirical formula (Barr et al., 2012; Bolton et al., 2004; Carey et al., 2010; Hayashi et
al., 2004). Thus, measuring all the water balance components of small-scale basins in thawing boreal peatland
complex remains challenging and unconstrained.
Here, we provide a multi-scale water balance analysis using field observations made in three small-scale
basins of a thawing boreal peatland complex in the headwater portion of Scotty Creek, a meso-scale, low-relief
basin near the southern permafrost limit in the Taiga Plains. The goal was to constrain the headwater sub-basin
water balances in a basin context. Specifically, our three objectives were to
(1) estimate daily sub-basin water losses (runoff, evapotranspiration), inputs (rainfall, snow water
equivalent) and storage change to quantify sub-basin water balances over three growing seasons (May-September,
2014-2016),

(2) examine sub-basin hydrological functioning in a basin context by comparing monthly sub-basin- and
corresponding basin water losses through evapotranspiration and runoff, and
(3) assess the long-term (1996-2022) annual basin water balance in relation to changes in land cover and
hydrological connectivity.





## 2   Methods

### 2.1   Study site

Our study site is within the headwater portion of the 130 (this study) to 202 km$^2$ (by Water Survey of Canada,
wateroffice.ec.gc.ca, last access: 31 May 2024) Scotty Creek basin (61°18'N, 121°18'W) situated 50 km south of
Fort Simpson, NT in the sporadic permafrost zone of the southern Taiga Plains (Fig. 1-a, b). The continental,
subarctic climate of the Fort Simpson region is characterized by long, cold winters and short, dry summers. Climate
normal (1981-2010) mean annual air temperature and mean annual total precipitation (P) are -2.8 °C and 388 mm,
respectively, of which 40 % falls as snow (Fort Simpson A, WMO ID: 71946, Environment and Climate Change
Canada, climate.weather.gc.ca, last access: 31 May 2024). No significant difference of snow water equivalent
(SWE) between Fort Simpson and observations made in the headwater portion of Scotty Creek were found,
suggesting that the Fort Simpson station is a good proxy of SWE for Scotty Creek (Connon et al., 2021). The snow-
covered season usually begins in mid- to late October and lasts until mid- to late April or early May. The snow-
covered season duration has shortened by 35 days between 1998 and 2014 (Chasmer and Hopkinson, 2017). It was
estimated that the permafrost loss rate across the basin has increased from 0.19 % year$^{-1}$ (1970-2000) to 0.58 %
year$^{-1}$ (2000-2015) since the 1970s (Chasmer and Hopkinson, 2017).
Underlain by various glacial tills, silts, and clays deposited during the last glacial retreat (Aylesworth and
Kettles, 2000), the relatively flat (average slope: 0.3 %; Quinton et al., 2003) study site is dominated by low-lying
peatland ecosystems with interspersed well-drained mineral uplands. The treed mineral uplands are covered by
trembling aspen (*Populus tremuloides*) and white spruce (*Picea glauca*). The low-lying peatland ecosystems
include spatially extensive treed permafrost peat plateaus ("forests"), and permafrost-free thermokarst landforms
including wetlands ("wetlands") and lakes (Fig. 1-c). Separated from the forests by narrow (a few meters), actively
thawing forest-wetland transition, the topographically lowered (0.5-1 m) lakes and wetlands receive some lateral
inflow from the surrounding forests. The wetlands occur mainly as saturated treeless collapse features and channel
fens (a few 10s m in width) route water to the Scotty Creek basin outlet and connect wetlands to the drainage
network (Quinton et al., 2019; Fig. 1-b).
The forest overstory is dominated by black spruce (*Picea mariana*) interspersed with tamarack (*Larix*
*laricina*). Forest understory and ground cover is dominated by birch shrubs (*Betula* spp.), bog Labrador tea
(*Rhododendron groenlandicum*), bog rosemary (*Andromeda polifolia*), reindeer lichen (*Cladina* spp.), feather moss
(*Pleurozium schreberi*) and *Sphagnum* spp., respectively (Garon-Labrecque et al., 2016). Abiotic conditions (e.g.,
soil moisture and temperature) change abruptly within a few meters across the transitions from 'drier and cooler'





forests to 'wetter and warmer' wetlands (Baltzer et al., 2014; Helbig et al., 2016c). Wetland vegetation in the
collapse features mostly includes *Sphagnum* spp. and ericaceous shrubs such as leatherleaf (*Chamaedaphne*
*calyculata*), and pod-grass (*Scheuchzeria palustris*) in the wettest sections. The channels are dominated by
herbaceous species including scattered tamarack and glandular birch (*Betula glandulosa*), abundant seaside
arrowgrass (*Triglochin maritima*) and bog buckbean (*Menyanthes trifoliata*), and some dense patches of
Cyperaceae species. Channel ground cover is dominated by woolly feathermoss (*Tomenthypnum nitens*) and ribbed
bog moss (*Aulacomnium palustre*).

Peat thickness across the headwater portion of Scotty Creek is generally >3 m and has a mean (± one

standard deviation, std) organic carbon (C) stock of $167 \pm 11$ kg C m$^{-2}$ (n = 3; Pelletier et al., 2017). Forest
permafrost thickness is <10 m (McClymont et al., 2013; Quinton et al., 2009) with a maximum active layer
thickness in late August/early September of <1 m, consistent with other studies (Desyatkin et al., 2021; Devoie et
al., 2021). Mid- to late growing season (June and late August/early September) wetland water table position (WTP)
usually ranges between 0.1 and 0.2 m below the ground surface, respectively (Helbig et al., 2016b). Table A1
shows a list of all variables and expressions used in this study, alongside the corresponding abbreviations.



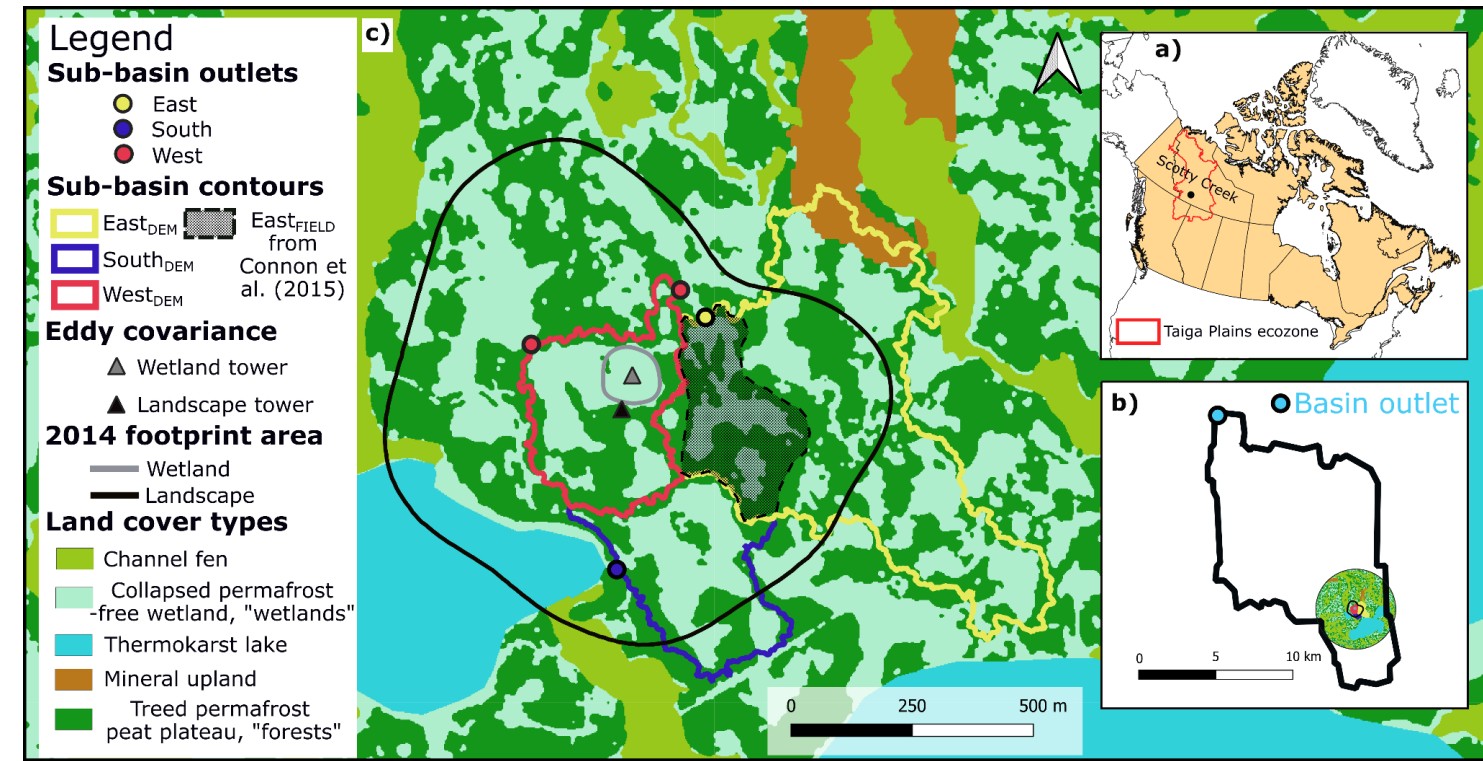

**Figure 1: a) Scotty Creek basin location in the southern Taiga Plains ecozone. b) Study site location within the Scotty Creek basin headwater portion. c) Landscape (i.e., boreal peatland complex) and wetland (i.e., collapsed permafrost-free wetland) towers 2014 eddy covariance footprint contours (90 % contribution) (Helbig et al., 2016b). Contours of the three small-scale basins, i.e., West, East, and South sub-basins, derived from automated terrain analysis using a DEM and of the East$_{FIELD}$ sub-basin derived from field observations (Connon et al., 2015). The land cover map is from Chasmer et al. (2014). The two outlets, South1 and South2, were located approximately 10 m apart, appearing as a single point. The basin (b) and the three sub-basins (c) water balances were studied over the periods 1996-2022 and 2014-2016, respectively.**



## 2.2 Sub-basin water balance: eddy covariance and supporting measurements

Boreal peatland complex ($ET_{LAND}$; 2014-2016) and wetland evapotranspiration ($ET_{WET}$; 2014-2016) were obtained from "nested" turbulent energy flux measurements using the eddy covariance technique (Baldocchi, 2014). Identical eddy covariance instrumentation was mounted at the top of a 15 m "landscape flux tower" (AmeriFlux-ID: CA-SCC) and at 1.9 m on a nearby (100 m) 2 m "wetland flux tower" (AmeriFlux-ID: CA-SCB; Fig. 1-c). The instrumentation on each tower included a three-dimensional sonic anemometer (CSAT3A; Campbell Scientific Inc., Logan, UT) and an open-path carbon dioxide ($CO_2$)/water vapor ($H_2O_{(g)}$) infrared gas analyzer (EC150) to measure the high-frequency fluctuations (10 Hz) in vertical wind velocity and sonic temperature, and $CO_2$ and $H_2O_{(g)}$ molar densities, respectively. Due to instrument failure, $CO_2$ and $H_2O_{(g)}$ molar densities were measured with an enclosed $CO_2/H_2O_{(g)}$ infrared gas analyzer (LI7200; LI-COR Biosciences Inc., Lincoln, NE) between March and August 2015. Further details on the instrumental set-up, the calibration and maintenance procedures, the data acquisition, processing and quality control, and the flux footprints calculation for the landscape and wetland towers are provided in Helbig et al. (2016c).

Supporting measurements on or near the landscape and wetland towers included incoming and outgoing short- and long-wave radiation (CNR4; Kipp & Zonen B.V., Delft, the Netherlands), rainfall (TR-525USW; Texas Instruments Inc., Dallas, TX), $T_{air}$ and relative humidity (HC2-S3; Rotronic AG, Basserdorf, Switzerland), soil temperature and moisture along vertical profiles, and relative wetland WTP (OTT PLS; OTT Hydromet GmbH, Kempten, Germany; Levelogger Gold F15/M5, Solinst Canada Ltd., Georgetown, ON; HOBO U20 Water Level Data Logger, Onset Computer Corporation, Bourne, MA). Wetland volumetric water content at 5 cm depth was measured with water content reflectometers (CS616; Campbell Scientific Inc.) at a wetland location in each of the three sub-basins. The different low-frequency ancillary data streams were stored as 30 min block averages in an external storage device connected to additional data loggers (CR1000, CR3000; Campbell Scientific Inc.). Forest and wetland SWE were obtained from snow depth (metal ruler) and density measurements (Eastern Snow Conference [30-cm$^2$ cross-sectional area] snow tube or snow sampler) along several representative forest and wetland transects during late March (i.e., late winter) snow surveys in 2014-2016 (Connon et al., 2015, 2021).

## 2.3 Sub-basin boundary delineation

The Scotty Creek basin headwater portion was studied using three small-scale basins ("sub-basins"): West (two outlets, West1 and West2), East (one outlet) and South (two outlets, South1 and South2), together draining ~48 % of the landscape flux footprint (Fig. 1-c). The wetland flux footprint area was located within the West sub-



basin. Delineating low-relief basin boundaries and thus drainage areas using automated terrain analysis techniques
remains challenging and estimates tend to vary depending on the level of topographic detail in the digital elevation
model (DEM) and the algorithm used (Al-Muqdadi and Merkel, 2011; Datta et al., 2022; Keys and Baade, 2019;
Moges et al., 2023). In boreal peatland complexes, differences between potential and effective drainage areas may
arise due to the presence of isolated wetlands disconnected from the drainage network and the basin outlet (Connon
et al., 2015).
We delineated the boundaries of potential drainage areas for the sub-basin outlets from a LiDAR derived
1 m DEM using terrain analysis techniques implemented in the ArcGIS Hydrology toolset from the Spatial Analyst
toolbox (version 10.2; Environmental Systems Research Institute, 2014; Chasmer et al., 2014). Considering the
low-relief landscape, we verified the resulting sub-basin boundaries plausibility (West$_{DEM}$, East$_{DEM}$, and South$_{DEM}$)
through visual interpretation of 2010 WorldView-2 imagery (Chasmer et al., 2014). Questionable boundary
sections were surveyed using a differential global positioning system (Leica SR530; Leica Geosystems, St. Gallen,
Switzerland) in post-processing kinematic mode (centimeter accuracy). Based on a decision-tree land cover
classification (Chasmer et al., 2014), West$_{DEM}$, East$_{DEM}$ and South$_{DEM}$ were dominated by forests (including forest-
wetland transitions) and wetlands (combined > 95 %). The resulting drainage areas and wetland-to-forest ratios are
0.105 km$^2$ (West$_{DEM}$), 0.328 km$^2$ (East$_{DEM}$) and 0.099 km$^2$ (South$_{DEM}$), and 1.06 (West$_{DEM}$), 0.84 (East$_{DEM}$) and 1.24
(South$_{DEM}$), respectively (Fig. 1-c).
Focusing on hydrological connections between individual wetlands and the sub-basin outlets, the
boundaries of effective drainage areas for the West and East sub-basins were delineated previously (West$_{FIELD}$ and
East$_{FIELD}$; Connon et al., 2015). These delineations were based on visual inspection of the same DEM and 2010
WorldView-2 imagery used in the potential drainage areas delineation described in the previous paragraph followed
by extensive field observations. Permafrost barriers and permafrost-free hydrological connections to channels
around and between wetlands and the sub-basin outlets were identified using a frost probe. All wetlands in the West
sub-basin were hydrologically well-connected to the drainage system, resulting in similar drainage area estimates
for West$_{FIELD}$ (0.090 km$^2$) and West$_{DEM}$ (0.105 km$^2$). In the East sub-basin, several isolated wetlands were not
connected to the drainage system, resulting in a fivefold smaller drainage area estimate for East$_{FIELD}$ (0.068 km$^2$)
compared to East$_{DEM}$ (0.328 km$^2$). We used both drainage area estimates for the East sub-basin, East$_{DEM}$ and
East$_{FIELD}$, to calculate sub-basin Q. The South sub-basin contained one individual wetland directly connected to the
two outlets (Fig. 1-c), thus we expect the difference between effective and potential drainage area to be negligible
(South$_{FIELD}$ ≈ South$_{DEM}$).



## 2.4 Sub-basin water balance: discharge measurements

We estimated daily discharge (L day$^{-1}$) as open water flow at five narrow (1-8 m in width) stream channel locations (= sub-basin outlets) in the landscape and wetland towers vicinity using rectangular cutthroat flumes (Fig. S1 and S2). The flumes were constructed following open-source design plans (Siddiqui et al., 1996; Skogerboe et al., 1972), and installed 0.8 m above the channel bottom on wooden damming structures to divert the flow of water through the flumes. Half-hour WTP was measured every 5 minutes and averaged and recorded every 30 minutes at each flume from April to late August/early September in 2014-2016 using vented pressure transducers (DCX-38 VG; Keller AG, Winterthur, Switzerland). Rating curves (n = 15, one per flume in 2014-2016) to convert WTP to half-hour discharge estimates were obtained from manual discharge and WTP measurements made during and shortly after the snowmelt period in late April to early May (spring freshet) and late May (baseflow) in 2014-2016, respectively. Gaps in the half-hour discharge time series were filled in two steps. First, half-hour WTP recorded at nearby upstream wetland locations within the respective sub-basin (Haynes et al., 2018) were used to construct monthly and growing season (May-September) proxy rating curves with non-gap-filled half-hour discharge for each flume in 2014-2016. Discharge gap-filled with wetland WTP represented 74.8, 6.5 and 13.1 % of data for the West, East and South sub-basins, respectively. These monthly rating curves were then used to gap-fill the half-hour discharge time series. Growing season rating curves were used in case of insufficiently strong monthly proxy rating curves. Second, any remaining gaps (13.6, 11.7 and 21.1 % of data for the West, East and South sub-basins, respectively) due to missing upstream relative wetland WTP were gap-filled using linear regression analysis based on a mean 2014-2016 growing season proxy rating curve. Gap-filled half-hour discharge was summed to obtain daily discharge for the three sub-basins, which was converted to daily sub-basin runoff ($Q_{WEST}$, $Q_{EAST-FIELD}$, $Q_{EAST}$, and $Q_{SOUTH}$; mm day$^{-1}$) using the corresponding effective (East$_{FIELD}$ only) and potential drainage areas (West$_{DEM}$, East$_{DEM}$ and South$_{DEM}$).

## 2.5 Basin water balance: data sets

We obtained several data sets for Scotty Creek spanning 27 hydrological years (October-September 1996-2022). Instantaneous discharge for the Scotty Creek basin outlet (Fig. 1-b) along the Liard Highway (61°24'N, 121°26'W) is publicly available (Scotty Creek at Highway No. 7, 10ED009; Water Survey of Canada, wateroffice.ec.gc.ca). Daily P (mm day$^{-1}$; R and SWE) are publicly available for the nearest weather station in Fort Simpson (Fort Simpson A, WMO ID: 71946, Environment and Climate Change Canada, climatedata.ca, last access: 31 May 2024). We obtained daily ET (mm day$^{-1}$) for Scotty Creek (21 hydrological years: October-September





2001-2022) from the Breathing Earth System Simulator (BESS; Jiang et al., 2016), a global biophysical model with
a spatial resolution of 0.05° (Fig. S3). We used the average value of ET (2002-2022 period) to calculate the 1996-
2001 water balance.

We delineated a drainage area for the Scotty Creek basin outlet from the publicly available 90 m DEM of

the Shuttle Radar Topography Mission (SRTM, Hole-filled SRTM for the globe Version 4; Jarvis et al., 2008)
using automated terrain analysis techniques implemented in the ArcGIS Hydrology toolset from the Spatial Analyst
toolbox (Environmental Systems Research Institute (ESRI), 2014). The terrain analysis derived potential drainage
area was 130 km$^2$, thus smaller than previously published drainage area estimates for the Scotty Creek basin outlet:
134 km$^2$ (Burd et al., 2018), 139 km$^2$ (Chasmer and Hopkinson, 2017), 150 km$^2$ (Quinton et al., 2004), 152 km$^2$
(Connon et al., 2014) and 202 km$^2$ (Water Survey of Canada). For reproducibility and methodological consistency
with the sub-basin drainage areas, BESS estimates of ET were averaged across Scotty Creek using the terrain
derived drainage area (130 km$^2$, this study). All data sets were temporally aggregated to monthly and annual
(hydrological year: October - September) runoff ($Q_{BASIN}$), precipitation ($P_{BASIN}$), SWE and rainfall ($SWE_{BASIN}$,
$R_{BASIN}$), and evapotranspiration ($ET_{BASIN}$). We used the lower (130 km$^2$, this study) and upper basin drainage area
estimates (202 km$^2$, Water Survey of Canada) to calculate $Q_{BASIN}$ ($Q_{BASIN\_130}$ and $Q_{BASIN\_202}$, respectively).
**2.6     Multi-scale water balance analysis**

We calculated monthly (mm month$^{-1}$; West sub-basin), growing season (mm period$^{-1}$; West, East and South

sub-basins denoted as subscripted "WEST", "EAST" and "SOUTH") and annual (hydrological year: October -
September, mm year$^{-1}$; Scotty Creek basin denoted as subscripted "BASIN") water balances as:

$R + SWE = ET + Q + \Delta S$,                                                                                      (1)

where $\Delta S$ is water storage change, rain (R) plus snow water equivalent (SWE) represent the total precipitation (P),
ET is evapotranspiration. Groundwater discharge from permafrost thaw was expected to be negligible in boreal
peatland complexes (Connon et al., 2014; Quinton et al., 2019).

For simplicity, we loosely defined the growing season as the May-September period when actual

measurements for all water balance components (Eq. 1) for the complete months were available. For example, the
wetland WTP measurements started in May because before then, the wells were frozen. The WTP was used to
calculate $\Delta S_{SUB\text{-}BASIN}$ as we assumed that Q occurs from forests to the topographically lower wetlands (Wright et
al., 2022). Therefore, we calculated $\Delta S_{SUB\text{-}BASIN}$ for the West, East, and South sub-basins based on wetland $\Delta S$





using the sub-basin specific wetland area coverage ($A_{WET}$). $\Delta S_{WET}$ was calculated based on saturated and
unsaturated peat layers using WTP variation, volumetric water content at 5 cm depth, and the peat porosity values
at 3 (0.92) and 15 cm (0.86) from Isabelle et al. (2018).

Precipitation ($P_{SUB-BASIN}$) including rainfall (R) and snow water equivalent (SWE) just before the snowmelt

period in late March (i.e., $SWE_{MAX}$) was obtained from rain gauge measurements ($R_{WEST} = R_{EAST} = R_{SOUTH}$), and
calculated as weighted mean for each sub-basins ($SWE_{MAX\_SUB-BASIN}$) according to sub-basin specific cover areas
(i.e., wetland [$A_{WET}$] and forest areal coverage [$A_{FOR}$]) and associated measured SWE (i.e., forest [$SWE_{MAX\_FOR}$]
and wetland SWE [$SWE_{MAX\_WET}$]), respectively:

$SWE_{MAX\_SUB-BASIN} = (A_{FOR}\times SWE_{MAX\_FOR} + A_{WET}\times SWE_{MAX\_WET})/A_{SUB-BASIN}$                    (2)

where $A_{SUB-BASIN}$ denotes the sub-basin area. We added $SWE_{SUB-BASIN}$ to rainfall in May as we assumed that the
main contribution of snow to the $P_{SUB-BASIN}$, and thus to the growing season and annual water balances, occurred
mainly through the complete snowpack melting.

Average energy balance closure fractions at the landscape and wetland towers were 0.70 (0.67, 0.72 0.72

from 2014 to 2016) and 0.67 (0.65, 0.69 and 0.68 from 2014 to 2016), respectively. To account for sensible (H; W
$m^{-2}$) and latent heat (LE; W $m^{-2}$) underestimation, we applied the closure fraction correction by preserving the
Bowen ratio (H $LE^{-1}$), to obtain the corrected LE (i.e., ET) (Barr et al., 2012; Isabelle et al., 2020). The closure
fraction correction was calculated using 30 min average fluxes for the months of July to September, when the most
complete energy flux data were available. Mean growing season forest and wetland flux footprint area contributions
to $ET_{LAND}$ (corresponding to $ET_{WEST}$) measured at the landscape tower were approximately 50 % each (Helbig et
al., 2017; Helbig et al., 2016b; Warren et al., 2018; Fig. 1-c). In contrast, the mean growing season footprint for
$ET_{WET}$ consisted solely of wetland surrounding the tower (Helbig et al., 2016b; Warren et al., 2018). For the South
and East sub-basins, we calculated forest ET ($ET_{FOR}$, Eq. 3) using $ET_{LAND}$ and $ET_{WET}$ as:

$ET_{FOR} = (ET_{LAND} - A_{WET}/A_{SUB-BASIN} \times ET_{WET}) / (A_{FOR}/A_{SUB-BASIN})$                    (3)


Evapotranspiration for the South and East sub-basins was calculated as weighted means as for $SWE_{SUB-BASIN}$

(Eq. 2). Sub-basin runoff ($Q_{SUB-BASIN}$) was obtained from daily discharge measurements and the corresponding
sub-basin areas.





Annual basin water balances (mm year$^{-1}$, Eq. 1) were calculated using temporally aggregated precipitation- ($P_{BASIN}$) and rain ($R_{BASIN}$) measurements from Fort Simpson (Fort Simpson A, WMO ID: 71946, Environment and Climate Change Canada, climatedata.ca, last access: 31 May 2024), with snow water equivalent ($SWE_{BASIN}$) simply calculated as $P_{BASIN}$ minus $R_{BASIN}$, and ET estimates from BESS ($ET_{BESS\_BASIN}$). The $\Delta S_{BASIN}$ was calculated as the difference between the water inputs ($P_{BASIN}$) and outputs ($Q_{BASIN}$ and $ET_{BESS\_BASIN}$) of Eq. 1. A positive value indicated an increase in water stored in the sub-basin, and reciprocally.

We compared growing season monthly $Q_{SUB\text{-}BASIN}$ and $ET_{SUB\text{-}BASIN}$, both calculated as the means of the corresponding West, East and South sub-basin estimates, with $Q_{BASIN}$ and $ET_{BESS\_BASIN}$, respectively, using ordinary least squares (OLS) regression analysis. $Q_{BASIN}$ and $Q_{SUB\text{-}BASIN}$ used for this comparison were obtained from the drainage area derived from automated terrain analysis of a DEM in this study. Similarly, we compared monthly $ET_{LAND}$ with headwater estimates from BESS ($ET_{BESS\_HEAD}$) using OLS regression analysis. The OLS regressions uncertainty was estimated using bootstrapping with 1000 iterations. The headwater estimate of ET from BESS was calculated as a five-pixel average of the pixel containing the landscape tower and its surrounding pixels (Fig. S3). We examined the annual (hydrological year: October-September) hydrological balance components, i.e., $Q_{BASIN}$, $P_{BASIN}$, $R_{BASIN}$, and $SWE_{BASIN}$ time series (1996-2022), and calculated the annual ratio of runoff to precipitation (the runoff ratio).

## 3 Results

### 3.1 Meteorological conditions

The annual mean $T_{air}$ of the Fort Simpson region over the three-year study period is in the range (2014, within one std) and higher (2015, 2016, ~+1 °C) than the 27-year mean (1996-2022, Table 1). The first year of the three-year study period (2014) is much drier (~-100 mm), with less snow and rainfall, compared to the two other years and to the 27-year study. The annual total P is lower (2016) or higher (2015) than the 27-year mean, but within one std. The snow cover period beginning and end are consistent throughout the three years at Scotty Creek.

**Table 1.** Annual mean air temperature ($T_{air}$), total precipitation (P), snow water equivalent (SWE) and rainfall (R) at Fort Simpson airport (Fort Simpson A, WMO ID: 71946, Environment and Climate Change Canada, climatedata.ca, last access: 31 May 2024), and the dates of snowmelt end and the start of a spatially continuous snow cover, and the snow-free season length at Scotty Creek.



| | $T_{air}$ (°C) | P (mm) | SWE (mm) | R (mm) | Snowmelt end | Snow cover start | Snow-free season (days) |
|---|---|---|---|---|---|---|---|
| **2014** | -2.7 | 215 | 81 | 134 | May 4th | October 13th | 162 |
| **2015** | -1.3 | 392 | 117 | 274 | May 9th | October 15th | 159 |
| **2016** | -1.0 | 301 | 126 | 175 | May 3rd | October 9th | 159 |
| **1996-2022** | -2.3 ± 0.9 (std) | 355 ± 68 | 112 ± 24 | 243 ± 63 | – | – | – |

### 3.2 Sub-basin growing season water balances

The West, East and South and sub-basins hydrographs are dominated by the spring freshet caused by the rapid snowpack melting starting in late April (Fig. 2-a, b, c). Each year, the peak in $Q_{SUB-BASIN}$ occurs within two to four days after the snowmelt period starts. For each sub-basin, the spring freshet (April-May) Q is the lowest in 2014 (15, 44, 27 and 130 mm for the West, South, East$_{DEM}$ and East$_{FIELD}$ sub-basins, respectively) and the highest in 2016 (83 and 104 mm for the West and South sub-basins, respectively), with intermediate values in 2015 (54 and 77 for the West and South sub-basins, respectively). The peak in Q of 12 mm day$^{-1}$ is observed for the South sub-basin (highest wetland-to-forest ratio) in 2016, coinciding with a heavy rainfall event (>30 mm) 10 days before the snowmelt period (Fig. 2-c). Spring freshet accounts for 99 and 100 %, 73 and 87 %, and 83 and 89 % of the April-September Q in 2014, 2015 and 2016, for the West and South sub-basins, respectively. In contrast, the spring freshet for the East sub-basin corresponds to 41 and 47 % of the April-September Q in 2014 and 2015, respectively. Once the spring freshet ceases, only the East sub-basin sustains continuous Q throughout the remainder of the growing season (baseflow) in 2014 (drier than normal conditions; Fig. 2-a). All three sub-basins sustain continuous Q post-spring freshet in 2015 (wetter than normal conditions) but not in 2016 (drier than normal conditions; data only for West and South sub-basins in 2016). All post-spring freshet variations in Q are in response to individual storm events, reaching rainfall amounts of up to 30 mm day$^{-1}$.





Over the study period, average $ET_{LAND}$ is $2.9 \pm 1.1$ mm day$^{-1}$ (ranging from 0.6 to 5.5 mm day$^{-1}$) and $ET_{WET}$
is $3.3 \pm 1.5$ mm day$^{-1}$ (ranging from 0.4 to 8.1 mm day$^{-1}$). The boreal peatland complex daily ET ($ET_{LAND} \approx$
$ET_{WEST}$) increases continuously from 0.3 mm day$^{-1}$ in early April to up to 2.5 mm day$^{-1}$ in late May in parallel with
the snowpack rapid melting. From late May until late September, the ET rate ranges between 2.0 and 4.0 mm day$^{-1}$
$^{1}$ for 50 % of the time (Fig. 2). The total ET from April to September is the lowest in 2014, 366 mm, where average
$T_{air}$ is 11.1 °C over this period. In contrast, the total ET and mean $T_{air}$ are similar in 2015 and 2016 (447 and 458
mm; 11.5 and 11.6 °C, respectively). Comparatively, $Q_{WEST}$ is 15, 75 and 101 mm from April to September of
2014, 2015 and 2016, respectively. Thus, $ET_{WEST}$ is approximately 24, 6 and 5 times greater than $Q_{WEST}$ for 2014,
2015 and 2016, respectively.
Differences in growing season (May-September) water input as $P_{SUB-BASIN}$ and combined losses ($ET_{SUB-BASIN}$
and $Q_{SUB-BASIN}$) ranges between -211 mm (net loss: 2016, South) and +21 mm (net gain: 2015, West), resulting in
measured $\Delta S_{SUB-BASIN}$ of similar magnitudes (-250 mm [2016, South] to +3 mm [2015, East]) among sub-basins
and years (Fig. 3-a, b, c; Table 2). However, the difference between water input and losses for $East_{FIELD}$ sub-basin
is -354 and -311 mm, in 2014 and 2015, respectively (Fig. 3-b).
Considering the variation of water storage, water balance residuals of Eq. 1 for the growing season are
positive for the West (114, 122 and 34 mm in 2014, 2015, and 2016) and South (38 mm in 2016) sub-basins (Fig.
3, Table 2). In contrast, $RES_{EAST}$ and $RES_{EAST-FIELD}$ are negative (-81, -30 mm and -287 and -285 mm, in 2014 and
2015, respectively). For the West sub-basin, we recorded all the water balance components throughout the three-
year study period, allowing us to compute its monthly growing season water balance.





**Figure 2: Basin and sub-basins hydrographs in a) 2014, b) 2015, and c) 2016.** Daily rainfall ($R_{EAST} = R_{SOUTH} = R_{WEST}$, mm day$^{-1}$), boreal peatland complex evapotranspiration ($ET_{LAND}$) approximately corresponding to ET from the West sub-basin ($ET_{LAND} \approx ET_{WEST}$, mm day$^{-1}$), Q (mm day$^{-1}$) from the Scotty Creek basin, and Q (mm day$^{-1}$) from the East, South, and West sub-basins approximately draining the landscape tower eddy covariance footprint area (Fig. 1c). East$_{DEM}$ and East$_{FIELD}$ drainage areas are used to compute the lower and upper Q range contours (DOY = day-of-year).





**Table 2.** Growing season water balances (May-September, 2014-2016, mm) for the West, East, East$_{FIELD}$ (using
the effective field derived sub-basin surface, Connon et al., 2015), and South sub-basins at Scotty Creek (Fig. 3):
snow water equivalent (SWE$_{WEST}$, SWE$_{EAST}$, SWE$_{EAST-FIELD}$ and SWE$_{SOUTH}$), rainfall (R$_{WEST}$ = R$_{EAST}$ = R$_{SOUTH}$),
evapotranspiration (ET$_{LAND}$ ≈ ET$_{WEST}$, ET$_{EAST}$, ET$_{EAST-FIELD}$ and ET$_{SOUTH}$), runoff obtained from discharge
measurements and potential drainage area delineated with automated terrain analysis using a DEM (Q$_{WEST}$, Q$_{EAST}$,
Q$_{EAST-FIELD}$ and Q$_{SOUTH}$), and water storage change (ΔS$_{WEST}$, ΔS$_{EAST}$, ΔS$_{EAST-FIELD}$ and ΔS$_{SOUTH}$). The water balance
residual (RES) results from Eq. 1. Relative wetland water table position (WTP) and discharge data to calculate
ΔS$_{SOUTH}$ and Q$_{EAST}$ are not available in 2014 and 2015 (not measured), and 2016 (instrument failure), respectively.

| | SWE$_{SUB-BASIN}$ (mm) | R (mm) | ET$_{SUB-BASIN}$ (mm) | Q$_{SUB-BASIN}$ (mm) | ΔS$_{SUB-BASIN}$ (mm) | RES$_{SUB-BASIN}$ (mm) |
|---|---|---|---|---|---|---|
| **WEST sub-basin** | | | | | | |
| 2014 | 102 | 208 | 351 | 4 | -159 | 114 |
| 2015 | 167 | 316 | 426 | 35 | -101 | 122 |
| 2016 | 128 | 220 | 442 | 66 | -195 | 34 |
| **EAST sub-basin** | | | | | | |
| 2014 | 102 | 208 | 353 | 65 | -27 | -81 |
| 2015 | 169 | 316 | 437 | 74 | 3 | -30 |
| 2016 | 128 | 220 | 443 | – | -75 | – |
| **EAST$_{FIELD}$ sub-basin** | | | | | | |
| 2014 | 103 | 208 | -303 | -311 | -16 | -287 |
| 2015 | 172 | 316 | -413 | -358 | 2 | -285 |
| 2016 | 129 | 220 | -415 | – | -43 | – |
| **SOUTH sub-basin** | | | | | | |





| 2014 | 102 | 208 | 384 | 25 | – | – |
|------|-----|-----|-----|-----|------|----|
| 2015 | 167 | 316 | 452 | 40 | – | – |
| 2016 | 128 | 220 | 460 | 100 | -250 | 38 |

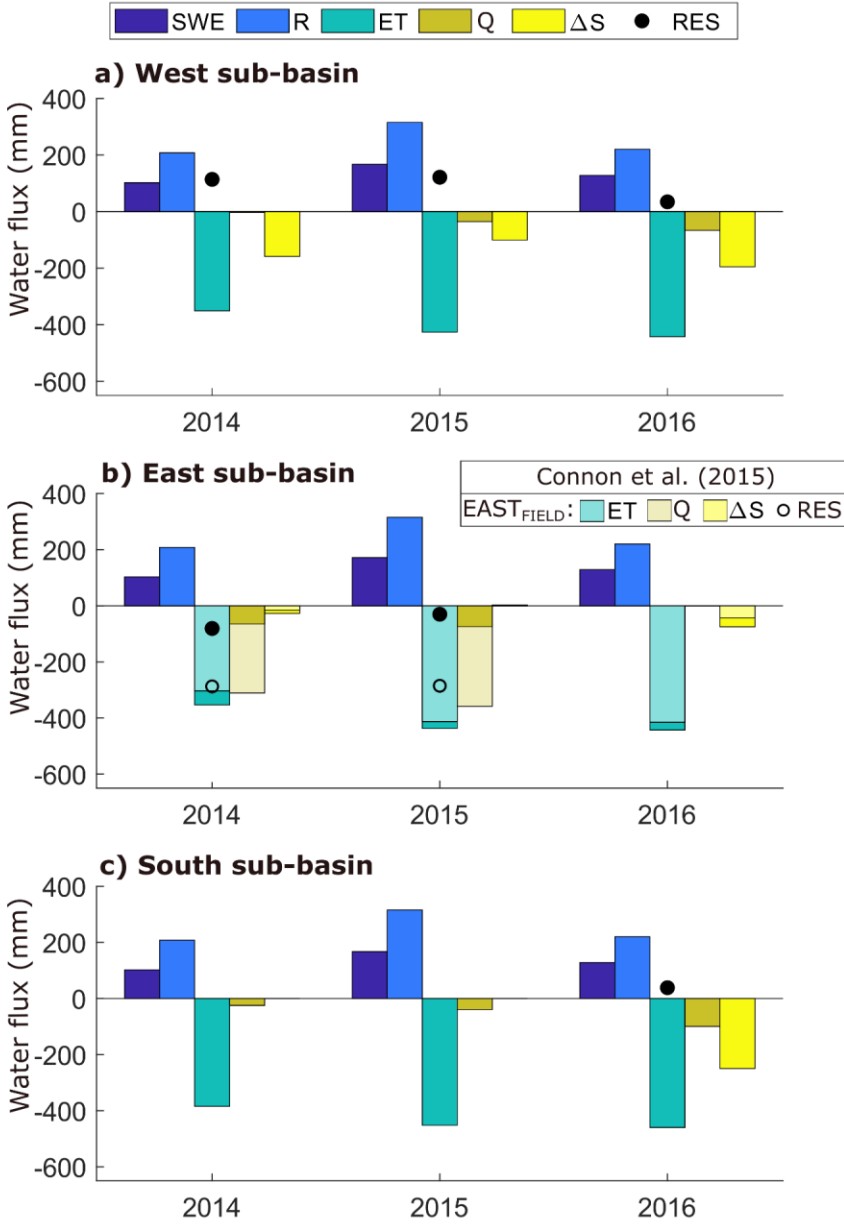



**Figure 3: Growing season (May-September, 2014-2016) water balances (mm) for the a) West, b) East and c) South sub-**
**basin: rainfall ($R_{EAST}$ = $R_{SOUTH}$ = $R_{WEST}$), snow water equivalent ($SWE_{EAST}$, $SWE_{SOUTH}$, and $SWE_{WEST}$),**
**evapotranspiration ($ET_{EAST}$, $ET_{SOUTH}$, and $ET_{LAND}$ ≈ $ET_{WEST}$), runoff derived from the terrain analysis drainage area**
**($Q_{EAST}$, $Q_{SOUTH}$, and $Q_{WEST}$), and water storage change ($\Delta S_{EAST}$, $\Delta S_{SOUTH}$, and $\Delta S_{WEST}$). The black dot symbol indicates**
**the water balance residual ($RES_{EAST}$, $RES_{SOUTH}$, and $RES_{WEST}$) resulting from Eq. 1. b) For the East sub-basin, $ET_{EAST-}$**
**$_{FIELD}$, $Q_{EAST-FIELD}$, and $\Delta S_{EAST-FIELD}$ are estimated from the effective drainage area derived from field observations**
**($EAST_{FIELD}$, Connon et al., 2015). $SWE_{EAST-FIELD}$ is similar to $SWE_{EAST}$. The white dot indicates $RES_{EAST-FIELD}$. Relative**
**wetland water table position (WTP) and discharge data to calculate $\Delta S_{SOUTH}$ and $Q_{EAST}$ are not available in 2014 and**
**2015 (not measured), and 2016 (instrument failure), respectively.**
### 3.3 Sub-basin monthly growing season water balance - West sub-basin
The negative $\Delta S_{WEST}$ in May indicates a large reduction in water stored in the West sub-basin, even though
total water input ($R_{WEST}$ plus $SWE_{WEST}$) exceeds by 20 (2016) to 50 % (2014 and 2015) water losses ($ET_{WEST}$ plus
$Q_{WEST}$, Fig. 4-a, b, C, Table 3). This discrepancy is reflected in the large positive monthly water balance residuals
($RES_{WEST}$) in May each year (149-, 176-, and 117- mm in 2014, 2015, and 2016, respectively), reaching almost
twice the magnitude of $\Delta S_{WEST}$ in 2014 and 2015 (Fig. 4-a, b). In contrast, monthly $RES_{WEST}$ from June to
September for all three years are an order of magnitude lower than those of May (-41 to 0 with a mean of -14 mm,
Table 3). In the three years, $ET_{WEST}$ is similar during the early- to mid-growing season (June to August: mean
monthly ± one std, $ET_{WEST}$ = 95 ± 9 mm). Mean monthly late growing season $ET_{WEST}$ in September is 45 ± 8 mm.
For the June to September period, 2014 total $R_{WEST}$ (188 mm) is lower than total $ET_{WEST}$ (291 mm) and $\Delta S_{WEST}$ is
-69 mm. Similarly, in 2016, $ET_{WEST}$ (361 mm) largely exceeds $R_{WEST}$ (185 mm) and $\Delta S_{WEST}$ is -110 mm. In contrast,
in June to September 2015, $R_{WEST}$ (291 mm) is close to $ET_{WEST}$ (336 mm) and $\Delta S_{WEST}$ is -10 mm.



Figure 4: Growing season monthly (May-September, 2014-2016) water balances (mm month$^{-1}$) for the West sub-basin: rainfall (R$_{WEST}$), snow water equivalent (SWE$_{WEST}$), evapotranspiration (ET$_{LAND}$) approximately corresponding to ET from the West sub-basin (ET$_{LAND}$ ≈ ET$_{WEST}$), runoff (Q$_{WEST}$), and water storage change (ΔS$_{WEST}$). The black dot symbol indicates the monthly water balance residual (RES$_{WEST}$) resulting from Eq. 1.





**Table 3.** Growing season monthly water balances (May-September, 2014-2016, mm month$^{-1}$) for the West sub-basin (Fig. 4): snow water equivalent (SWE$_{WEST}$), rainfall (R$_{WEST}$), boreal peatland complex evapotranspiration (ET$_{LAND}$) approximately corresponding to ET from the West sub-basin (ET$_{LAND}$ ≈ ET$_{WEST}$), runoff (Q$_{WEST}$), and water storage change ($\Delta$S$_{WEST}$). RES$_{WEST}$ indicates the monthly water balance residual resulting from Eq. 1.

| 2014 | SWE$_{WEST}$ (mm) | R$_{WEST}$ (mm) | ET$_{WEST}$ (mm) | Q$_{WEST}$ (mm) | $\Delta$S$_{WEST}$ (mm) | RES$_{WEST}$ (mm) |
|---|---|---|---|---|---|---|
| **2014** | | | | | | |
| MAY | 102 | 20 | 60 | 3 | -90 | 149 |
| JUN | 0 | 64 | 85 | 0 | -8 | -14 |
| JUL | 0 | 52 | 94 | 0 | -38 | -4 |
| AUG | 0 | 57 | 76 | 0 | -1 | -17 |
| SEP | 0 | 15 | 36 | 0 | -21 | 0 |
| **2015** | | | | | | |
| MAY | 167 | 24 | 90 | 15 | -90 | 176 |
| JUN | 0 | 43 | 114 | 4 | -64 | -12 |
| JUL | 0 | 146 | 94 | 4 | 63 | -15 |
| AUG | 0 | 47 | 83 | 6 | -19 | -23 |
| SEP | 0 | 55 | 45 | 6 | 9 | -5 |
| **2016** | | | | | | |
| MAY | 128 | 36 | 82 | 50 | -85 | 117 |
| JUN | 0 | 60 | 119 | 11 | -54 | -16 |
| JUL | 0 | 45 | 107 | 2 | -58 | -6 |
| AUG | 0 | 29 | 82 | 1 | -35 | -19 |
| SEP | 0 | 51 | 53 | 2 | 37 | -41 |





### 3.4    Comparison between sub-basin and basin evapotranspiration and runoff

Comparable spring freshet peaks are observed between the basin and sub-basins, except for the driest year (2014), when Q in the basin hydrograph (<0.6 mm) is substantially lower than in the sub-basins (from 1.6 to 9.4 mm; Fig. 2). The spring freshet contributions (April-May) to Q over the April-September period at the basin scale varies between 50 to 79 % over the period 2014-2016, i.e., in the range observed for the three sub-basins (from 41 to 100 %). The monthly Q comparison (using drainage area obtained with terrain analysis techniques) between the sub-basins and the basin is coherent. The greatest absolute difference is twofold in May (from 1.6 to 2.3; Fig. 5-a). Total ET from the BESS model over the April-September period ranges from 237 to 252 mm for both basin and headwater portion while values measured from the landscape tower range from 366 (2014) to 458 mm (2016). Consistently, the monthly comparison of ET shows lower values from the modeled ET (BESS) at both basin and headwater scales compared to the measured one with the flux towers (Fig. 5-b, c). Higher water losses ($\Delta S_{SUB\text{-}BASIN}$) in 2014 and 2016 observed for the growing season sub-basins (Fig. 3) are consistent with the annual (hydrological year: October-September) basin response $\Delta S_{BASIN}$ (Fig. 6-a), which we will present in detail over the long-term (1996-2022) in the following section.

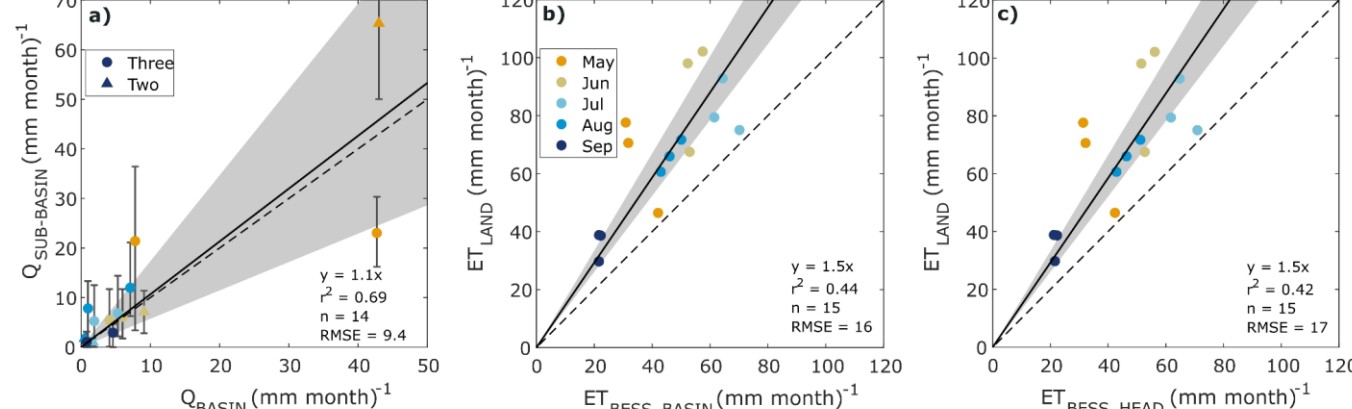

**Figure 5: Monthly comparison of growing season (May-September 2014 – 2016, mm month$^{-1}$) water losses (ET and Q) between the Scotty Creek basin (x-axis) and the sub-basins located in the headwater portion (y-axis). a) Q$_{BASIN}$ and average (vertical error bar corresponding to minimum and maximum) Q estimates for the East, South and West sub-basins (Q$_{SUB-BASIN}$). Q are obtained from the drainage area derived from automated terrain analysis using a DEM. Symbol shape indicate the number of sub-basin months available to calculate sub-basins mean Q. No discharge data to calculate Q$_{SUB-BASIN}$ is available in September 2016. b) Basin ET estimates with the BESS (ET$_{BESS\_BASIN}$) and c) headwater ET estimates from the BESS (ET$_{BESS\_HEAD}$) with corresponding (y-axis) landscape tower eddy covariance measurements (ET$_{LAND}$), respectively. For a), b) and c), the continuous black line is the ordinary least square (OLS) regression. The OLS regression uncertainty (grey colored-band) is estimated using bootstrapping with 1000 iterations. The stippled black line is the 1:1-line.**





### 3.5  Basin annual water balance

Over the 27-year (1996-2022) study period, annual water inputs are dominated by rainfall, ranging from 111 to 324 mm (mean ± std, 243 ± 63 mm) while $SWE_{BASIN}$ ranges from 81 to 181 mm (mean ± std, 112 ± 24 mm, Fig. 6-a, Table S1). For water outputs, annual ET estimated with BESS ranges between 223 to 311 mm (mean ± std, 261 ± 22 mm) over the 2002-2022 period (Fig. 6-a). For the period 2002-2022, annual $Q_{BASIN\_130}$ and $Q_{BASIN\_202}$ range from 26 to 317 mm (mean ± std = 164 ± 81 mm) and from 17 to 204 mm (mean ± std = 105 ± 52 mm), respectively. Thus, annual ET is between 2.2 and 3.5 times higher than annual Q, given the range of drainage area values.

$ET_{BESS\_BASIN}$ and $SWE_{BASIN}$ are relatively stable over time (261 ± 22 mm, 112 ± 24 mm, respectively, Fig. 6-a). $\Delta S_{BASIN}$, $R_{BASIN}$ and $Q_{BASIN}$ experience higher between-year variability from 1996 to 2022 ($\Delta S_{BASIN\_130}$: -60 ± 75 mm, $\Delta S_{BASIN\_202}$: -5 ± 63 mm; R: 243 ± 63 mm; $Q_{BASIN\_130}$: 155 ± 76 mm; $Q_{BASIN\_202}$: 100 ± 49 mm) than $ET_{BESS\_BASIN}$ and $SWE_{BASIN}$.

$\Delta S_{BASIN\_202}$ and $\Delta S_{BASIN\_130}$ range from -172 to 105 mm and from -95 to +121 mm, respectively. $\Delta S_{BASIN}$ decreases from ~+120 to 0 mm over 1996 to 2001 while Q exhibits an increase from ~30 to ~140 mm. $\Delta S_{BASIN}$ is negative (~100 mm) over the 2004-2014 period. Then, $\Delta S_{BASIN}$ is alternatively positive and negative from 2015 to 2022 for both drainage areas (Fig. 6-a).

The annual ratio of runoff to precipitation (i.e., the runoff ratio, Fig. 6-b) ranges from 0.1 to 0.5 (runoff ratio $_{202}$) and from 0.1 to 0.8 (runoff ratio $_{130}$). Runoff ratio strongly increases from 1996 to 2002 (from ~0.1 to 0.4-0.6, runoff ratio $_{202}$ and runoff ratio $_{130}$ average is 0.2 and 0.3, respectively) followed by a period of higher and more stable values until 2012 (runoff ratio $_{202}$ and runoff ratio $_{130}$ average for 2003-2012 are 0.4 and 0.6, respectively). For the 2013-2022 period, the runoff ratio is more variable but on average lower (runoff ratio $_{202}$ = 0.2 and runoff ratio $_{130}$ = 0.4) than for the period 2003-2012.





**Figure 6: a) Annual (hydrological year: October-September, 1996-2022) water balances (mm year$^{-1}$) for the Scotty Creek basin obtained from daily precipitation (P$_{BASIN}$) and rainfall measurements (R$_{BASIN}$) resulting in snow water equivalent (SWE$_{BASIN}$ = P$_{BASIN}$ - R$_{BASIN}$), daily runoff (Q$_{BASIN}$), evapotranspiration (ET) estimates from the BESS (ET$_{BESS\_BASIN}$). ET for the 1996-2001 period (dashed green line) corresponds to the 2002-2022 average period. Basin-scale water storage change (ΔS$_{BASIN}$) is the difference between incoming and outgoing water fluxes. b) Annual ratio of runoff to precipitation (i.e., the runoff ratio). Hashed area corresponds to the average runoff ratio over the temporal period considered (1996-2002; 2003-2012; 2013-2022). For subplots a) and b), the range of values for Q$_{BASIN}$, ΔS$_{BASIN}$ and runoff ratio corresponds to the lowest and highest basin drainage areas, i.e., 130 and 202 km².**





## 4    Discussion

### 4.1    Growing season water balance components in three small-scale basins of a boreal peatland complex: Objective 1

From mid May until the end of September, the growing season water balances are dominated by water inputs and losses through R and ET. Growing season daily ET ranges among values commonly observed elsewhere across the boreal biome with higher wetland than forest ET (Arain et al., 2003; Isabelle et al., 2018; Nakai et al., 2013; Volik et al., 2021; Wu et al., 2010). Higher wetland ($2.9 \pm 1$ mm day$^{-1}$) than forest ET ($1.7 \pm 0.6$ mm day$^{-1}$) is reported at Scotty Creek in June-mid July 2013 and transpiration from black spruce accounts only for approximately 1-2 % of forest ET (Warren et al., 2018).

The spring freshet extends through April and May and dominates water losses from the three small-scale basins. The spring freshet contribution to growing season water losses is the lowest for the East sub-basin. Despite the East sub-basin drainage area uncertainty, the range of wetland-to-forest ratio for the East sub-basin (0.34 to 0.84) is lower than for the two other sub-basins (South: 1.24 and West: 1.06). The greater forest area in the could lead to more post-spring freshet Q, as the gradually deepening frost table can promote subsurface Q (Sjöberg et al., 2021). In contrast, during the mid-growing season, wetlands can act as 'gatekeepers' reducing hydrological connectivity (Connon et al., 2015; Phillips et al., 2011). Land cover control over Q dynamics in other permafrost affected basins are observed in, for example, a mountainous permafrost area where differences in vegetation types affected the rainfall-runoff relationship (Genxu et al., 2012).

Regarding the monthly water balance, the high residuals observed in May over the three years (Fig. 4-a, b, c) might be explained by the inclusion of snowmelt input (i.e., SWE) in May. It was not possible to obtain snowmelt rates and timing and wells to measure wetland WTP are frozen in April. Thus, SWE$_{MAX}$ estimated just before snowmelt in late March is included in the May water balance, highlighting the difficulty in integrating this crucial period of high discharge into the growing season balance. Despite the observational challenges, particular attention should be paid to this winter-to-spring transitional period, which is profoundly influenced by climate change. Firstly, the spring freshet is shown to occur earlier (Chasmer and Hopkinson, 2017; Mack et al., 2021; Woo et al., 2008). Predictions of Q for the 2040-2069 and 2070-2099 periods in a tundra site in the Canadian Arctic show an advance in snowmelt timing on average up to 25 days compared to the 1961-1990 climate normal (Pohl et al., 2007). Secondly, earlier snowmelt start leads to a longer snowmelt period, as projected for the Liard River watershed, resulting in a more gradual snowmelt (Woo et al., 2008). The winter-to-spring transitional period is projected to increase in northeastern North America by +15 to +28 days by the end of the century, signifying an





ongoing hydrological shift (Grogan et al., 2020). Except for May, the remainder of the growing season shows a closed water balance with very low residuals (Fig. 4), suggesting that obtaining water storage from measured wetland WTP and water content is appropriate in low-relief landscapes such as the thawing boreal peatland complex in this study. To better understand the hydrological functioning from small- to meso-scale basins, we compare hydrographs and monthly average Q and ET from the three headwater sub-basins with those obtained at the basin scale, as described in the following section.

## 4.2 Small-scale basin evapotranspiration and runoff from a boreal peatland complex in a meso-scale basin context: Objective 2

Similarly to the growing season sub-basin water balances, the annual basin water balance (Fig. 6) has higher water losses in 2014 and 2016 (Fig. 3). In addition, we observed that with two independent data sets (i.e., sub-basins measurements from this study and basin publicly available data), ET is the major annual (on average more than twice as high) water loss at both sub-basin and basin scales. The hydrographs at both scales are comparable, i.e., dominated by the spring freshet peak, typical of regions with a subarctic nival regime (Gandois et al., 2021; Woo et al., 2008). However, an exception occurs during the driest year (2014) when the basin Q peak is more than 10 times lower than for sub-basins (Fig. 2). This difference might be partially explained by the higher proportional coverage of wetlands in the headwater sub-basins (~40 %) compared to the entire basin (~20 %; Chasmer et al., 2014). The quantity of water stored in saturated wetlands is expected to be higher than in mineral uplands (McCarter et al., 2020; Price, 1987), thus potentially sustaining a higher runoff ratio in case of small SWE in years such as 2014.

For the concurrent monitoring period at both scales (2014-2016), sub-basin Q agrees well with basin-scale Q (Fig. 5-a). May exhibits the highest difference (twofold difference), highlighting difficulties in capturing the spring freshet, a period of flooding, snow damming, or lateral outflows, adversely affecting discharge measurement accuracy. In addition, the discrepancy in Q between the sub-basin and basin scales may also be attributed to time lag effects. Spring freshet peak is delayed (~2-4 days) between the headwater sub-basins and the basin outlet (Fig. 2). Thus, a Q portion occurring in late April within the headwater sub-basins might be accounted for in May at the basin scale. The observed Q difference in May could also be partially explained by comparing basin (i.e., 130 and 202 km²) and sub-basin scale (<1 km²) snow depth and melt dynamics, as the snowpack can be heterogeneous in forests and melt faster in wetlands than in forest (Connon et al., 2021; Nousu et al., 2024).

Our results also indicates that annual modeled ET (BESS) used at the basin scale underestimated (~100 mm) observed ET (Fig. 5-b). Given that the wetland ET is higher than forest ET (Warren et al., 2018), ET





underestimation from BESS can be explained by the land cover heterogeneity at the basin scale. The northern, i.e.,
downstream, portion of the basin is dominated by mineral upland areas that are better drained and mainly covered
by deciduous or mixed forest stands (Chasmer et al., 2014). Modeled ET is lower than measured ET at the sub-
basin scale, probably underestimating the contribution of wetlands (Fig. 5-c). This difference might be attributed
to the tendency of BESS to underestimate the spatial variability of ET in wetland areas (Jiang and Ryu, 2016).

**4.3 Annual basin water balance in relation to changes in land cover and hydrological connectivity: Objective 3**

Several lines of evidence show how increased rates of thaw have resulted in permafrost loss at Scotty Creek
over the past few decades (Baltzer et al., 2014; Chasmer and Hopkinson, 2017; Helbig et al., 2016a; Quinton et al.,
2019). For example, air photographs (1970, 1977, 2000) and LiDAR derived DEM comparison over the Scotty
Creek basin headwater portion suggest an abrupt increase in permafrost loss rate from 0.19 % year$^{-1}$ (of total basin
area) between 1970 and 2000 to 0.58 % year$^{-1}$ from 2000 to 2015 triggered by a strong El Nino Southern Oscillation
(ENSO) phase in 1997/1998 (Chasmer and Hopkinson, 2017). Time series analysis of P at Fort Simpson and $Q_{BASIN}$
at Scotty Creek (1973-2015) identified a significant increase in annual runoff ratio after 1998, which cannot be
solely attributed to P (Chasmer and Hopkinson, 2017). This increase is associated with accelerated permafrost thaw
and especially the thaw of permafrost ridges, acting as barriers, allowing the hyrological connection of isolated
wetlands to the drainage system (Connon et al., 2014, 2015; Haynes et al., 2018; Quinton et al., 2019). Similarly,
an increase in annual Q (1996-2012) in four meso-scale basins draining into the Liard River (including Scotty
Creek) is attributed to an increase in hydrological connectivity, specifically the hydrological connections between
wetlands that result from the loss of permafrost barriers (Connon et al., 2014). This period of increase in Q is
interpreted as a transient period, estimated to be on the order of years to decades (Haynes et al., 2018). The strong
increase in runoff ratio observed in our work, from ~0.1 to 0.4-0.6 over the period 1996 to 2002 followed by a
stable and high runoff ratio (2003-2012) suggests that Q had increased without an increase in P (Fig. 6-b). In
addition to Scotty Creek, the seven other thawing peatland dominated basins in the Taiga Plains studied by Mack
et al. (2021) (i.e., Martin, Jean Marie, Birch, Trout, Blackstone, Hay and Keg) have experienced increases in the
runoff ratio over 1970-2016. Permafrost plays a role in Q generation by limiting water storage (Connon et al., 2014,
2015; Wright et al., 2009). For example, Carey et al. (2010) show that permafrost presence reduces storage and
enhances Q, ultimately leading to a high runoff ratio compared to permafrost-free basins with similar P. However,
in the Boreal Plains ecozone, south of the Taiga Plains, runoff ratio in 20 low-relief meso-scale basins over 25
years is significantly and positively correlated to peatland cover for mesic and wet periods (Devito et al., 2023).



Consistently, peatlands favor near surface saturation and Q generation (Devito et al., 2017; Devito et al., 2023). In
the Taiga Plains, permafrost loss, although increasing the active layer thickness, is therefore not opposite to an
increase in runoff ratio because of the increase in saturated wetlands hydrologically connected to the drainage
network.

Beyond the short-term and transient increase in runoff ratio in basins dominated by thawing boreal peatland

complexes, understanding their long-term Q dynamics remains challenging due to strong ecohydrological
feedbacks (Shirley et al., 2022; Song et al., 2024; Walvoord and Kurylyk, 2016). In the Scotty Creek basin
headwater portion, Haynes et al. (2022) estimate a -1.4 % forest loss between 2010 and 2018. The resulting increase
in hydrological connectivity lead to additional permanent Q and transient Q increased through drainage of
connected wetlands (Connon et al., 2014; Haynes et al., 2018). However, we observed a decrease in Q from 2009
to 2019 (Fig. 6-a) and the average runoff ratio over the period 2013-2022 decreases compared to the period 2003-
2012 (Fig. 6-b). Near surface peat layer high hydraulic conductivity favors drainage compared to deeper peat layers
(Ingram, 1978; Morris et al., 2011; Quinton et al., 2008). Therefore, the decrease in runoff ratio observed after
2012 could be due to a decrease in drainage efficiency (i.e., decrease in ΔS). A decrease in drainage efficiency is
compatible with wetland drying, limiting the near surface and more permeable peat layers saturation. This
interpretation is supported by the documented drying of hydrologically connected wetlands at Scotty Creek (Haynes
et al., 2018), allowing the development of hummocks over 2010-2018 (Haynes et al., 2022). Plant communities
succession at a 10 year horizon after wetland formation, leading to peat growth above the water table, can also
participate in limiting water saturation of upper layers (Errington et al., 2024). In addition to land cover,
precipitation regimes can contribute to the changes in runoff ratio. The peak in runoff ratio occurring in 2020 (0.5-
0.8) might be explained by the rainiest year recorded between 1996 and 2022 (Fig. 6-b). During wet conditions,
ephemerally connected wetlands can increase the effective drainage area (Connon et al., 2015) and dry periods can
decrease the runoff ratio by disconnecting some wetlands from the drainage network. Thus, long-term monitoring
would help to disentangle the effects of P and land cover changes on the runoff ratio in this rapidly changing
environment.

Change in land cover will also impact ET, potentially exerting a considerable influence on both the water

balance and the regional climate. Boreal biome wetlands ET is higher than forests at midday during the growing
season (Helbig et al., 2020a). 21$^{st}$ century projected wetland ET exceeds forest ET by more than 20 % in
approximately one-third (Representative Concentration Pathways [RCP] 4.5 scenario) and two-thirds (RCP 8.5
scenario) of the boreal biome (Helbig et al., 2020b). Thus, long-term measured and modeled ET comparisons





remain necessary since ET can play a crucial role in the future water balance of boreal peatland complexes near the
southern permafrost limit.

### 4.4   Effective versus potential drainage area: implications for water balance studies

Defining basin and sub-basin boundaries and drainage areas in low-relief landscapes such as vast swaths
Taiga Plains using automated terrain analysis techniques is challenging and estimates tend to vary, at least partly,
depending on the DEM used (Al-Muqdadi and Merkel, 2011; Datta et al., 2022; Keys and Baade, 2019; Moges et
al., 2023). Although difficult to apply across large regions, field observations are crucial in low-relief landscapes
for defining the effective drainage area (Connon et al., 2015). Our comparison of effective and potential drainage
areas, from field observations and automated terrain analysis of a DEM showed that both estimates are consistent
for the sub-basin almost exclusively composed of connected wetlands (factor 1.2, West sub-basin, Fig. 1-c).
However, the two drainage areas exhibit important differences for the sub-basin with a high proportion of isolated
wetlands (fivefold, East sub-basin). Not surprisingly, the potential drainage area derived is higher than the effective
drainage area. Field observations may lead to a more precise delineation of the effective drainage area contributing
to the drainage system (Connon et al., 2015). However, regarding the growing season water balance for the East
sub-basin (Fig. 3-b), the water balance residual is 3.5 to 9.5 higher using the effective drainage area. In this case,
the automated terrain analysis derived drainage area is more adequate to close the water balance. Subsurface water
flows can occur at greater depths in permafrost-free basins (Sjöberg et al., 2021). Unobserved subsurface flows,
such as through taliks, defined as perennially thawed ground below the active layer (Devoie et al., 2019), potentially
lead to an underestimation of the effective drainage areas from field observations.
At the basin scale, automated terrain analysis produces different drainage areas (Burd et al., 2018; Chasmer
and Hopkinson, 2017; Connon et al., 2014; Quinton et al., 2004; Water Survey of Canada) with the two most
distinct estimates being used in this study (i.e., 130 and 202 km²). The increase in wetlands hydrologically
connected to the effective drainage area due to permafrost thaw is expected to be captured by the substantial
increase in runoff ratio from 1996 to 2012 (Fig. 5). Delineating drainage areas at sub-basin and basin scales remains
a challenge, with proportionally larger errors in smaller areas such as the East, West and South sub-basins. Thus,
minor differences in landscape heterogeneity (e.g., wetland connectivity to the drainage system) may lead to large
variation in the drainage area. Since the southern permafrost limit undergoes rapid permafrost thaw and associated
land cover change (Quinton et al., 2019), better constraining the hydrological connectivity of small low-relief basins
can help in quantifying and modeling water and carbon losses (Gao et al., 2018; Wei et al., 2024).


## 4.5 Constraining water balance in thawing boreal peatland complexes: broader implications and perspectives

In this study we examined the hydrological functioning of a thawing boreal peatland complex near the southern permafrost limit expected to move northwards in the decades to come (Smith et al., 2022). Rapid changes in both atmospheric conditions (e.g., P and $T_{air}$) and ground thermal regimes underscore the need for reinforcing and continuing long-term hydrological monitoring to observe trends, identify breaks in time series, and understand changes in hydrological processes (Chasmer et al., 2017; Laudon et al., 2017; Tetzlaff et al., 2017).

Non-linear hydrological responses (e.g., runoff ratio, ET, hydrological connectivity, WTP) to changes in P and increased rates of permafrost thaw are associated with changes in soil physical properties, microbial communities and vegetation, collectively impacting local (e.g., subsistence activities), regional (e.g., weather) and global ecosystem services such as carbon storage (e.g., the net ecosystem carbon balance [NECB]; Camill et al., 2001; Chapin et al., 2006; Ernakovich et al., 2022; Jones et al., 2022; Li et al., 2023; Shirley et al., 2022). Assessing if thawing boreal peatland complexes are a net source or sink of carbon, once vertical and lateral fluxes are considered is therefore an important avenue of research (Song et al., 2024). For example, a recent review showed that dissolved organic carbon concentration can be elevated in sporadic and discontinuous permafrost areas and tend to increase with permafrost thaw (Heffernan et al., 2024). Thus, understanding of Q producing mechanisms such as the spring freshet is essential for quantifying lateral carbon exports towards the NECB (Chapin et al., 2006; Gandois et al., 2021; Laudon et al., 2004).

Long-term hydrological monitoring may also help in understanding how gradual changes (e.g., vegetation shift, increasing $T_{air}$) are interlinked with more frequent and intense disturbances (e.g., weather extremes, abrupt permafrost thaw, wildfires) (Li et al., 2023). Wildfires are shown to accelerate permafrost thaw (Gibson et al., 2018), an increasing issue for ecosystem services. 2023 is a record-breaking year especially in terms of surface burned across Canada (MacCarthy et al., 2024; Wang et al., 2024). As water table position and moisture can constitute an indicator of fire risk, understanding the water balance dynamics of peatland dominated basins may help in managing fire risk (Kartiwa et al., 2023; Mortelmans et al., 2024). At Scotty Creek, the site experienced a fire in October 2022. While the wetland flux tower and some flume boxes are still intact (Fig. S4), the landscape flux tower was rebuilt after the fire in March 2023. As our work helps to elucidate the hydrological functioning of a rapidly thawing boreal peatland complex, it can serve as an initial baseline for understanding the combined effects of permafrost thaw accelerated by wildfire.



## 5    Conclusions

This study contributes to a better understanding of the hydrological functioning of small-scale basins (i.e., sub-basins) within the headwater portion of a meso-scale basin (i.e., basin) in the Taiga Plains in western Canada. Our key findings are:

- Determining Q in low-relief landscapes such as thawing boreal peatland complexes is challenging because
    - sub-basin and basin boundaries and resulting drainage areas must be approached with caution since permafrost ridges act as barriers isolating wetlands from the effective drainage system, and
    - of difficulties in integrating the spring freshet into the growing season water balance.
- The small-scale headwater portion is representative of the corresponding meso-scale basin. At both scales, our analysis shows that
    - ET is the dominating water loss, on average more than twice than Q,
    - growing season (sub-basin) and annual water balance components temporal dynamics (basin) are similar,
    - spring freshet peaks are similar, except for the driest year, when basin Q is more than ten times lower than sub-basin Q, and
    - spring freshet contributions to the April-September Q are similar.
- Over the long-term (1996-2022), basin scale runoff ratio changes are partly attributed to change in land cover and associated hydrological connectivity. While the increase of runoff ratio is attributed to increased hydrological connectivity and wetland drainage (1996 to 2002), the stabilization (2003 to 2012) and decrease in runoff ratio (2013 to 2022) raise questions about the respective roles of changes in land cover and precipitation regimes.





## 6 Appendices

**Table A1.** List of all variables and expressions used in this study (left column), alongside the corresponding abbreviations (right column).

| Spatial information | |
|---|---|
| $A\text{-}_{WET,\ FOR}$ and $_{SUB\text{-}BASIN}$ | Wetland, forest and sub-basin area. |
| Basin | Meso-scale basin, $10^1$-$10^3$ km². In this study, this refers to the Scotty Creek basin (drainage area estimates from 130 to 202 km²). |
| DEM | Digital elevation model. |
| East-$_{FIELD}$ | East sub-basin drainage area derived from field observations (Connon et al., 2015). |
| Forest | Treed permafrost peat plateau. |
| Sub-basin | Small-scale basin, $<10^1$ km². In this study, the three small-scale basins are headwater sub-basins, called South, West and East, within the Scotty Creek meso-scale basin, see Fig. 1. |
| Wetland | Collapsed permafrost-free wetland. |
| Wetland-to-forest ratio | Ratio of wetland area to forest area. |
| West-, East-, and South-$_{DEM}$ | Sub-basin drainage area derived from automated terrain analysis using a digital elevation model (DEM). |
| Temporal information | |





| | |
|---|---|
| Growing season | The period from May to September over which the sub-basin water balances are calculated. |
| Spring freshet | Late April to early May runoff peak from snowmelt. |
| 27-year study | The period from 1996 to 2022 over which the annual basin water balance is calculated (hydrological year: October to September, 1995-10 to 2022-09). |
| **Hydrological variables** | |
| ET | Evapotranspiration. |
| $ET_{BESS\_HEAD}$ | Headwater portion ET modeled with BESS (Breathing Earth System Simulator). |
| $ET_{BESS\_BASIN}$ | Basin ET modeled with BESS. |
| $ET_{FOR}$ | ET calculated from $ET_{LAND}$ and $ET_{WET}$, see Eq. 3. |
| $ET_{LAND}$ | ET measured at the landscape flux tower. |
| $ET_{WET}$ | ET measured at the wetland flux tower. |
| P | Precipitation. |
| Q | Runoff. |
| R | Rainfall. |
| RES | Water balance residual resulting from Eq. 1. |
| Runoff ratio | Ratio of runoff to precipitation. |



| | |
|---|---|
| SWE, $SWE_{MAX}$ | Snow Water Equivalent. Maximum Snow Water Equivalent just before the snowmelt period in late March, see Eq. 2. |
| WTP | Water Table Position. |
| $\Delta S$ | Water storage change. |
| ET-, P-, Q-, R-, SWE-, $\Delta S_{BASIN,\ BASIN\_130}$ and $_{BASIN\_202}$ | Basin water balance components. _130 and _202 specify the drainage area in km². |
| ET-, P-, Q-, R-, SWE-, $\Delta S_{WEST}$, -$_{EAST}$ and -$_{SOUTH}$ | Water balance component for the corresponding sub-basin. |
| ET-, P-, Q-, R-, SWE-, $\Delta S_{EAST-FIELD}$ | Water balance component for the East sub-basin with the drainage area derived from field observations (Connon et al., 2015). |
| ET-, P-, Q-, R-, SWE-, $\Delta S_{SUB-BASIN}$ | Water balance component for the sub-basins. |
| **Environmental variables, acronyms** | |
| NECB | Net Ecosystem Carbon Balance. |
| RCP | Representative Concentration Pathways. |
| SRTM | Shuttle Radar Topography Mission. |
| Std | Standard deviation. |
| $T_{air}$ | Air Temperature. |

698



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



## 8 Code and data availability

Additional data are provided to this work as Supplementary Material. Further information can be supplied on request to the corresponding author.

## 9 Author contribution

**AL:** formal analysis, writing – original draft, writing – review and editing, **GHG:** formal analysis, data curation, methodology, writing – original draft, writing – review and editing, **MH:** data curation, writing – review and editing, **JF:** writing – review and editing, **YR:** data curation, writing – review and editing, **MD:** writing – review and editing, **RC:** data collection and instrumentation, formal analysis, writing – review and editing, **WQ:** formal analysis, writing – review and editing, **TM:** writing – review and editing, **OS**: Conceptualization; formal analysis; data curation, funding acquisition; methodology; supervision; writing – original draft; writing – review and editing.

## 10 Competing interests

The authors declare that they have no conflict of interest.

## 11 Acknowledgements

We gratefully acknowledge the support of the Dehcho First Nations, particularly the Liidlii Kue First Nation, for their support of our research activities on their traditional land. OS acknowledges support through TED Audacious for Permafrost Pathways, the Canada Research Chair (CRC-2018-279 00259), NSERC Discovery Grants (DGPIN-280 2018-05743) and FQRNT Projet de Recherche en Équipe programs (RQ000082), and the Global Water Futures project Northern Water Futures. This work also benefited from ArcticNet funding for the Dehcho Collaborative on Permafrost (*DCoP*). This research is part of Can-Peat: Canadian peatlands as nature-based climate solutions (https://uwaterloo.ca/can-peat). This project was undertaken with the financial support of the Government of Canada. Ce projet a été réalisé avec l'appui financier du gouvernement du Canada.