# Peer review of "Multi-scale water balance analysis of a thawing boreal peatland complex near the southern permafrost limit in northwestern Canada"

_Hydrology and Earth System Sciences, 2024_

## Author Comment (AC1)

**Multi-scale water balance analysis of a thawing boreal peatland complex near the southern permafrost limit in western Canada**

Alexandre Lhosmot1\*, Gabriel Hould Gosselin1,2\*, Manuel Helbig1,3, Julien Fouché1,4, Youngryel Ryu5, Matteo Detto6, Ryan Connon7, William Quinton8, Tim Moore9 and Oliver Sonnentag1, 10

1Département de géographie, Université de Montréal, Montréal, QC, Canada

2Department of Geography and Environmental Sciences, Northumbria University, Newcastle upon Tyne, UK

3Department of Physics & Atmospheric Science, Dalhousie University, Halifax, NS, Canada

4LISAH, Université de Montpellier, INRAE, IRD, Institut Agro, AgroParisTech, Montpellier, France

5Department of Landscape Architecture and Rural Systems Engineering, Seoul National University, Seoul, South Korea

6Department of Ecology and Evolutionary Biology, Princeton University, Princeton, NJ, USA

7Environment and Climate Change, Government of the Northwest Territories, Yellowknife, NT, Canada

8Cold Regions Research Centre, Wilfrid Laurier University, Waterloo, ON, Canada

9Department of Geography, McGill University, Montréal, QC, Canada

10Department of Geography and Environmental Studies, Wilfrid Laurier University, Waterloo, ON, Canada

\*These authors share the co-first authorship.

*Correspondence to:* Alexandre Lhosmot (alexandrelhosmot@gmail.com) and Oliver Sonnentag (oliver.sonnentag@umontreal.ca)

**Supplementary figures**

---

## Author Comment (AC2)

**Multi-scale water balance analysis of a thawing boreal peatland complex near the southern permafrost limit in western Canada**

Alexandre Lhosmot[1*], Gabriel Hould Gosselin[1,2*], Manuel Helbig[1,3], Julien Fouché[1,4], Youngryel Ryu[5], Matteo Detto[6], Ryan Connon[7], William Quinton[8], Tim Moore[9] and Oliver Sonnentag[1, 10]

[1]Département de géographie, Université de Montréal, Montréal, QC, Canada
[2]Department of Geography and Environmental Sciences, Northumbria University, Newcastle upon Tyne, UK
[3]Department of Physics & Atmospheric Science, Dalhousie University, Halifax, NS, Canada
[4]LISAH, Université de Montpellier, INRAE, IRD, Institut Agro, AgroParisTech, Montpellier, France
[5]Department of Landscape Architecture and Rural Systems Engineering, Seoul National University, Seoul, South Korea
[6]Department of Ecology and Evolutionary Biology, Princeton University, Princeton, NJ, USA
[7]Environment and Climate Change, Government of the Northwest Territories, Yellowknife, NT, Canada
[8]Cold Regions Research Centre, Wilfrid Laurier University, Waterloo, ON, Canada
[9]Department of Geography, McGill University, Montréal, QC, Canada
[10]Department of Geography and Environmental Studies, Wilfrid Laurier University, Waterloo, ON, Canada

*These authors share the co-first authorship.

*Correspondence to:* Alexandre Lhosmot (alexandrelhosmot@gmail.com) and Oliver Sonnentag (oliver.sonnentag@umontreal.ca)

**Supplementary figures**

[Figure]

**Figures S1:** Cutthroat flume installed at the West sub-basin outlet West1 in a) 2014 (i.e., before the late-season wildfire in October 2022), and b) 2024 (i.e., after the late-season wildfire in October 2022).

[Figure]

**Figure S2:** Location and description of the pixels (0.05 degrees of resolution) used for estimating ET with the Breathing Earth System Simulator (BESS) model (Jiang et al., 2016). The landscape map was produced by Chasmer et al. (2014). The black contour corresponds to the basin boundaries derived-from the SRTM DEM (130 km²).

[Figure]

**Figure S3:** Cumulative snowmelt evolution estimates for 2014, 2015, and 2016. The coloured line represents the median of the 10,000 simulations, while the shaded area represents the interquartile range (25th to 75th percentiles). The vertical black dashed line marks May 1st. We performed a simple snowmelt model using the temperature index equation (Pomeroy and Brun, 2001; Fontrodona-Bach et al., 2025):

Snowmelt (mm of water equivalent) = $C\_f \times (T\_air - T\_threshold)$

where, $C\_f$ is the melt factor (mm °C$^{-1}$ day$^{-1}$), $T\_air$ is the daily mean air temperature measured at Scotty Creek, $T\_threshold$ is the threshold temperature at which snow begins to melt. $T\_threshold$ ranged from -1 °C to +1 °C (Fontrodona-Bach et al., 2025 [preprint], and references therein). $C\_f$ was estimated using the relationship between snow density and $C\_f$ described by Rango and Martinec (1995). Given that snow density at Scotty Creek ranges from 0.11 to 0.29 (Connon et al., 2021), the corresponding $C\_f$ values were estimated to range from 1 mm °C$^{-1}$ day$^{-1}$ to 3 mm °C$^{-1}$ day$^{-1}$. To account for these uncertainties, we performed 10,000 simulations using a Monte Carlo approach, where $C\_f$ and $T\_thresholds$ were randomly sampled within their respective ranges.

References:

Connon, R.F., Chasmer, L., Haughton, E., Helbig, M., Hopkinson, C., Sonnentag, O., Quinton, W.L., 2021. The implications of permafrost thaw and land cover change on snow water equivalent accumulation, melt and runoff in discontinuous permafrost peatlands. Hydrological Processes 35, e14363. https://doi.org/10.1002/hyp.14363

Fontrodona-Bach, A., Schaefli, B., Woods, R., Larsen, J.R., 2025. [Preprint] Estimating robust melt factors and temperature thresholds for snow modelling across the Northern Hemisphere. https://doi.org/10.5194/egusphere-2025-1214

Rango, A., Martinec, J., 1995. Revisiting the degree-day method for snowmelt computations. J American Water Resour Assoc 31, 657–669. https://doi.org/10.1111/j.1752-1688.1995.tb03392.x

[Figure]

**Figure S4:** Composite hydrographs for 14-year periods of 1995-2008 and 2009-2022 at Scotty Creek basin outlet. Colored bands correspond to 95 % confidence intervals. The inlet shows a zoom on the spring freshet onset.

[Figure]

**Figure S5:** Annual, spring freshet (April–May), and summer (June–September) runoff ratios at Scotty Creek basin. Late-winter snow water equivalent is included in the spring freshet runoff ratio calculation. The dashed vertical gray line corresponds to 2012.

[Figure]

**Figure S6:** Difference between spring freshet and summer runoff ratios.

[Figure]

**Figure S7:** Annual runoff vs. a) current-year effective precipitation, b) previous year's effective precipitation, and c) cumulative effective precipitation from current and previous years. Effective precipitation is defined as precipitation minus evapotranspiration.

[Figure]

**Figure S8:** Cross-correlation plot between annual basin runoff and effective precipitation (precipitation minus evapotranspiration).

**Table S1.** Growing season water balances (May-September, 2014-2016, mm) for the West, East, East$_{FIELD}$ (using the effective field derived sub-basin drainage area, Connon et al., 2015), and South sub-basins at Scotty Creek (Figure 3): snow water equivalent (SWE$_{WEST}$, SWE$_{EAST}$, SWE$_{EAST-FIELD}$ and SWE$_{SOUTH}$), rainfall (R$_{WEST}$ = R$_{EAST}$ = R$_{SOUTH}$), evapotranspiration (ET$_{LAND}$ ≈ ET$_{WEST}$, ET$_{EAST}$, ET$_{EAST-FIELD}$ and ET$_{SOUTH}$), runoff obtained from discharge measurements (Q$_{WEST}$, Q$_{EAST}$, Q$_{EAST-FIELD}$ and Q$_{SOUTH}$) and water storage change (ΔS$_{WEST}$, ΔS$_{EAST}$, ΔS$_{EAST-FIELD}$ and ΔS$_{SOUTH}$). Potential drainage area was delineated with automated terrain analysis using a digital elevation model (Q$_{WEST}$, Q$_{EAST}$ and Q$_{SOUTH}$) and derived from field observations (Q$_{EAST-FIELD}$). The water balance residual (RES) results from Eq. 1. Wetland water table position (WTP) and discharge data to calculate ΔS$_{SOUTH}$ and Q$_{EAST}$ are not available in 2014 and 2015 (not measured), and 2016 (instrument failure), respectively.

| | SWE$_{SUB-BASIN}$ (mm) | R (mm) | ET$_{SUB-BASIN}$ (mm) | Q$_{SUB-BASIN}$ (mm) | ΔS$_{SUB-BASIN}$ (mm) | RES$_{SUB-BASIN}$ (mm) |
|---|---|---|---|---|---|---|
| **WEST sub-basin** | | | | | | |
| 2014 | 102 | 208 | 351 | 4 | -159 | 114 |
| 2015 | 167 | 316 | 426 | 35 | -101 | 122 |
| 2016 | 128 | 220 | 442 | 66 | -195 | 34 |
| **EAST sub-basin** | | | | | | |
| 2014 | 102 | 208 | 353 | 65 | -27 | -81 |
| 2015 | 169 | 316 | 437 | 74 | 3 | -30 |
| 2016 | 128 | 220 | 443 | – | -75 | – |
| **EAST$_{FIELD}$ sub-basin** | | | | | | |
| 2014 | 103 | 208 | 303 | 311 | -16 | -287 |
| 2015 | 172 | 316 | 413 | 358 | 2 | -285 |
| 2016 | 129 | 220 | 415 | – | -43 | – |
| **SOUTH sub-basin** | | | | | | |
| 2014 | 102 | 208 | 384 | 25 | – | – |
| 2015 | 167 | 316 | 452 | 40 | – | – |
| 2016 | 128 | 220 | 460 | 100 | -250 | 38 |

**Table S2.** Growing season monthly (May-September, 2014-2016) water balances (mm month$^{-1}$) for the West sub-basin (Figure 4): snow water equivalent (SWE$_{WEST}$), rainfall (R$_{WEST}$), boreal peatland complex evapotranspiration (ET$_{LAND}$) approximately corresponding to ET from the West sub-basin (ET$_{LAND}$ ≈ ET$_{WEST}$), runoff (Q$_{WEST}$), and water storage change (ΔS$_{WEST}$). RES$_{WEST}$ indicates the monthly water balance residual resulting from Eq. 1.

| 2014 | SWE$_{WEST}$ (mm) | R$_{WEST}$ (mm) | ET$_{WEST}$ (mm) | Q$_{WEST}$ (mm) | ΔS$_{WEST}$ (mm) | RES$_{WEST}$ (mm) |
|---|---|---|---|---|---|---|
| **2014** | | | | | | |
| **MAY** | 102 | 20 | 60 | 3 | -90 | 149 |
| **JUN** | 0 | 64 | 85 | 0 | -8 | -14 |
| **JUL** | 0 | 52 | 94 | 0 | -38 | -4 |
| **AUG** | 0 | 57 | 76 | 0 | -1 | -17 |
| **SEP** | 0 | 15 | 36 | 0 | -21 | 0 |
| **2015** | | | | | | |
| **MAY** | 167 | 24 | 90 | 15 | -90 | 176 |
| **JUN** | 0 | 43 | 114 | 4 | -64 | -12 |
| **JUL** | 0 | 146 | 94 | 4 | 63 | -15 |
| **AUG** | 0 | 47 | 83 | 6 | -19 | -23 |
| **SEP** | 0 | 55 | 45 | 6 | 9 | -5 |
| **2016** | | | | | | |
| **MAY** | 128 | 36 | 82 | 50 | -85 | 117 |
| **JUN** | 0 | 60 | 119 | 11 | -54 | -16 |
| **JUL** | 0 | 45 | 107 | 2 | -58 | -6 |
| **AUG** | 0 | 29 | 82 | 1 | -35 | -19 |
| **SEP** | 0 | 51 | 53 | 2 | 37 | -41 |

**Table S3.** Annual (hydrological year [HY]: October-September) water flux components (mm year$^{-1}$) over the period 1996-2022: snow water equivalent (SWE), rainfall (R), precipitation (P), evapotranspiration (ET) from the Breathing Earth System Simulator (BESS) at the basin and basin's headwater portion scales (ET$_{BESS\_BASIN}$ and ET$_{BESS\_HEAD}$), runoff (Q$_{BASIN\_130}$ and Q$_{BASIN\_202}$), water storage change calculated from the other water flux components ($\Delta$S$_{BASIN\_130}$ and $\Delta$S$_{BASIN\_202}$) and runoff ratio (Runoff ratio$_{130}$ and Runoff ratio$_{202}$). The indices "130" and "202" indicate the surface of the basin in square kilometers.

| HY | SWE | R | P | ET BESS_BASIN | ET BESS_HEAD | Q BASIN_130 | Q BASIN_202 | ΔS BASIN_130 | ΔS BASIN_202 | Runoff ratio_130 | Runoff ratio_202 |
|---|---|---|---|---|---|---|---|---|---|---|---|
| 1995-10 / 1996-09 | 107 | 305 | 412 | – | – | 46 | 30 | 105 | 122 | 0.1 | 0.1 |
| 1996-10 / 1997-09 | 91 | 304 | 395 | – | – | 103 | 67 | 31 | 68 | 0.3 | 0.2 |
| 1997-10 / 1998-09 | 95 | 306 | 401 | – | – | 115 | 74 | 24 | 65 | 0.3 | 0.2 |
| 1998-10 / 1999-09 | 126 | 283 | 409 | – | – | 123 | 79 | 25 | 69 | 0.3 | 0.2 |
| 1999-10 / 2000-09 | 135 | 297 | 431 | – | – | 163 | 105 | 8 | 66 | 0.4 | 0.2 |
| 2000-10 / 2001-09 | 107 | 324 | 431 | – | – | 188 | 121 | -18 | 49 | 0.4 | 0.3 |
| 2001-10 / 2002-09 | 181 | 233 | 413 | 262 | 255 | 231 | 149 | -79 | 3 | 0.6 | 0.4 |
| 2002-10 / 2003-09 | 119 | 285 | 404 | 253 | 250 | 159 | 103 | -8 | 48 | 0.4 | 0.3 |
| 2003-10 / 2004-09 | 96 | 151 | 246 | 267 | 258 | 110 | 71 | -131 | -92 | 0.4 | 0.3 |
| 2004-10 / 2005-09 | 139 | 215 | 354 | 272 | 270 | 160 | 103 | -78 | -21 | 0.5 | 0.3 |
| 2005-10 / 2006-09 | 128 | 211 | 339 | 268 | 268 | 241 | 155 | -170 | -84 | 0.7 | 0.5 |
| 2006-10 / 2007-09 | 116 | 234 | 349 | 270 | 270 | 191 | 123 | -112 | -44 | 0.5 | 0.4 |

| | | | | | | | | | | |
|---|---|---|---|---|---|---|---|---|---|---|
| 2007-10 / 2008-09 | 132 | 199 | 331 | 269 | 267 | 155 | 100 | -94 | -38 | 0.5 | 0.3 |
| 2008-10 / 2009-09 | 140 | 311 | 450 | 286 | 282 | 305 | 196 | -140 | -32 | 0.7 | 0.4 |
| 2009-10 / 2010-09 | 82 | 301 | 384 | 285 | 282 | 270 | 174 | -172 | -76 | 0.7 | 0.5 |
| 2010-10 / 2011-09 | 133 | 277 | 410 | 311 | 306 | 227 | 146 | -128 | -47 | 0.6 | 0.4 |
| 2011-10 / 2012-09 | 81 | 253 | 334 | 280 | 278 | 210 | 135 | -156 | -81 | 0.6 | 0.4 |
| 2012-10 / 2013-09 | 106 | 153 | 259 | 269 | 264 | 71 | 45 | -81 | -56 | 0.3 | 0.2 |
| 2013-10 / 2014-09 | 81 | 134 | 215 | 272 | 271 | 26 | 17 | -83 | -73 | 0.1 | 0.1 |
| 2014-10 / 2015-09 | 118 | 274 | 392 | 254 | 254 | 135 | 87 | 2 | 51 | 0.3 | 0.2 |
| 2015-10 / 2016-09 | 126 | 175 | 301 | 258 | 257 | 108 | 70 | -65 | -27 | 0.4 | 0.2 |
| 2016-10 / 2017-09 | 84 | 196 | 280 | 233 | 235 | 91 | 58 | -43 | -11 | 0.3 | 0.2 |
| 2017-10 / 2018-09 | 90 | 179 | 269 | 223 | 222 | 75 | 48 | -29 | -2 | 0.3 | 0.2 |
| 2018-10 / 2019-09 | 84 | 246 | 330 | 240 | 237 | 63 | 41 | 27 | 50 | 0.2 | 0.1 |
| 2019-10 / 2020-09 | 96 | 321 | 416 | 236 | 232 | 317 | 204 | -137 | -24 | 0.8 | 0.5 |
| 2020-10 / 2021-09 | 101 | 298 | 399 | 230 | 230 | 151 | 97 | 18 | 72 | 0.4 | 0.2 |
| 2021-10 / 2022-09 | 127 | 111 | 238 | 239 | 244 | 145 | 93 | -147 | -95 | 0.6 | 0.4 |

Updated Figure 2:

[Figure]

Updated Figure 6:

---

## Author Response (AR1)

Dear associate editor,

We are thankful for the supportive and constructive comments you and two reviewers provided for our manuscript. We thoroughly revised the manuscript based on the review, and hope that our efforts, with the insight from your end, have improved the manuscript.

Below, you will find the original comments of Reviewer 1 and 2 in bold, followed by our detailed responses and descriptions of revisions to the manuscript.

Kind regards,

Alexandre Lhosmot, on behalf of all authors
* * *
**REVIEWER 1**

**The paper by Lhosmot and co-authors presents considerable empirical data from the well studied Scotty Creek watershed where research has been ongoing for decades. In this work, the authors collate water balance data across temporal and spatial scales to answer questions regarding hydrological partitioning and dominant flux, and also touch on issues related to basin delineation, data products versus observations and other issues.**

**First, I want to acknowledge that there is a lot of field data that has gone into this work and it is important to acknowledge the true challenges that this entails. The method section provides sufficient detail for the readers to evaluate where this data came from, how it was processed, and points to companion papers for further information. In short, I see no substantive issues with the data presented or the approaches used to assemble it.**

Thanks.

**The paper is quite long because of all the data presented. Data is presented in both tabular and figure format directly, and I'd suggest the author to select one for the main part of the manuscript and move the other to the SI. Perhaps the figures as opposed to the tables.**

We agree with the reviewer. We moved Tables 2 and 3 to the Supplementary Material as Tables S1 and S2, respectively.

**While water balances are interesting for those looking at long-term studies impacts of hydrological change, the most important contributions come from the insights of the authors. My question to the authors is: what new insights are there in this paper? For me, the discussion section does not advance much on the results, restating general patterns without a critical assessment of what insights can be gained from this data set. The general fluxes patterns are not new, but is there something that can be gained by linking this small scale with larger scale data? What here advances our understanding of process from this well studied system?**

This study provides key insights into how the hydrological responses of rapidly thawing boreal peatland complexes–at both sub-basin and basin scales–are shaped by complex factors (e.g., hydrological connectivity) that extend beyond year-to-year changes in precipitation and evapotranspiration (ET). Further details are provided in response to the following reviewer's comment.

**In addition, much is made of the changes in runoff ratio at the basin scale, but again, this is speculative at the moment talking about changes in drainage efficiency, changing plant communities, etc. Can the authors convince us that this is the case. The runoff ratios are presented on an annual basis, so I'm unsure if that plant community and peat drying hypothesis holds. Is the change in the ratio largely due to differences in the freshet SWE to Q in May? Or is it the rest of the year? These are important issues considering that these changes in runoff ratios are more regional. These ecosystems are changing remarkably fast, but how can something as complex as hydrological response (and runoff which integrates a lot of processes) be linked to that. I would have liked some additional analysis here, perhaps looking at previous season wetness or fall wetness which may influence the following spring runoff. There is talk of regional teleconnections that at this long-term scale could be explored even simply.**

We conducted two additional analyses to address this comment and set of questions. Specifically, we examined (1) runoff ratios during different periods of the year, and (2) current-year and previous-year influences on effective precipitation (precipitation minus ET) on annual runoff.

(1) We calculated runoff ratios for the spring freshet (using April-May rainfall and the late-winter SWE) and summer (June-September). Both periods exhibit an increasing trend from 1996 to 2011 (Figure 1), reinforcing that runoff increased independently of precipitation changes. Runoff ratios during spring freshet and summer periods show two distinct phases: 1996-2011/2012 with an overall increase in runoff ratio, and post-2012 with generally lower and more variable runoff ratios (Figure 1). Similar runoff ratios between spring freshet and summer periods suggest that a common driver or set of drivers, likely hydrological

connectivity and drainage efficiency, influences both spring freshet and summer runoff patterns.

[Figure]

Figure 1. Annual, spring freshet (April–May), and summer (June–September) runoff ratios at Scotty Creek basin. The late winter SWE is included in the spring freshet runoff ratio calculation. The dashed vertical gray line corresponds to 2012.

We also examined the difference between spring freshet (April-May) and summer (June-September) runoff ratios (Figure 2). Runoff ratio differences for most years remain within the narrow range of -0.1 to +0.1. Before 2012, only 2008 stands out, but more pronounced run ratio differences became more frequent after 2012. The generally greater decline in summer runoff ratio compared to spring freshet suggests pronounced wetland drying during summer months, likely caused by their temporary disconnection from the effective drainage area (Connon et al., 2015). From 2010 to 2018, wetland drying associated with hummock development has been described previously in the headwater portion of the Scotty Creek basin (Haynes et al., 2020, 2022). Such widespread wetland drying may help explain the overall decrease in runoff ratio at the basin scale observed since 2012.

[Figure]

Figure 2. Difference between spring freshet and summer runoff ratios.

(2): To assess the role of the previous year's hydrological conditions on annual runoff, we created scatter plots of annual basin runoff (hydrological year, October to September) as a function of:

(a) current year's effective precipitation,

(b) previous year's effective precipitation,

(c) cumulative current and previous year's effective precipitation.

Effective precipitation is defined here as the difference between precipitation (late winter SWE and rainfall) and ET from BESS for a hydrological year. Our analysis suggests that current year's effective precipitation provides the best linear correlation with runoff, while previous year's effective precipitation has no explanatory power ($R^2$ = 0.0, Figure 3).

[Figure]

Figure 3. Annual runoff vs. (a) current year's effective precipitation, (b) previous year's effective precipitation, and (c) cumulative effective precipitation from current and previous years.

However, correlation between current-year effective precipitation and runoff remains weak ($R^2$ = 0.2), suggesting that other processes, such as changes in landscape hydrological connectivity, might play a more important role in controlling runoff. Additionally, cross-correlation analysis reveals that previous years' effective precipitation cannot explain basin runoff (Figure 4). The best correlation (Pearson's correlation = 0.45) was with the current year's effective precipitation.

[Figure]

Figure 4. Cross-correlation plot between annual basin runoff and effective precipitation.

Figure 1, 2, 3 and 4 have been added to the Supplementary Material as Figure S5, S6, S7 and S8, respectively. Additional analyses (1) and (2) led to modifications in the revised version of the manuscript:

Starting at line 48 (original submission) and 49 (revised submission) (Abstract): "
[revised manuscript text omitted]

**Section 4.5 is perhaps not needed. Much of this has been touched on in the introduction to the paper so the authors may consider trimming or deleting. I'm not sure how much value it has.**

We shortened this section by removing its first paragraph (L647 to 651, original submission) which was more general and somewhat redundant with the Introduction section. We clarified the remaining part of Section 4.5 as follows:

Starting at line 652 (original submission) and 664 (revised submission): "Non-linear hydrological responses such as changes in runoff ratio, ET and water table position to variations in precipitation and hydrological connectivity driven to by permafrost thaw are linked to shifts in soil physical properties, microbial communities and vegetation composition and structure. These interconnected changes collectively influence ecosystem services at multiple scales, including local (e.g., subsistence activities), regional (e.g., water storage) and global levels (e.g., carbon storage as reflected in the net ecosystem carbon balance [NECB]; Camill et al., 2001; Chapin et al., 2006; Ernakovich et al., 2022; Jones et al., 2022; Li et al., 2023; Shirley et al., 2022). Assessing whether thawing boreal peatland complexes act as a net source or sink of carbon (NECB), once both vertical and lateral fluxes are considered, is therefore an important avenue of research (Song et al., 2024). For example, a recent review showed that dissolved organic carbon concentration can be elevated in sporadic and discontinuous permafrost areas and tend to increase with permafrost thaw (Heffernan et al., 2024). Thus, understanding the mechanisms driving runoff, such as the spring freshet, is essential for quantifying lateral carbon exports to NECB (Chapin et al., 2006; Gandois et al., 2021; Laudon et al., 2004).

Long-term hydrological monitoring is also essential for understanding how gradual changes (e.g., vegetation shift, increasing Tair) are interlinked with more frequent and intense pulse disturbance events (e.g., weather extremes, abrupt permafrost thaw, wildfires) (Li et al., 2023). Wildfires have been shown to accelerate permafrost thaw (Gibson et al., 2018), posing an increasing threat to ecosystem services. The year 2023 set a record for surface burned across Canada (MacCarthy et al., 2024; Wang et al., 2024). As water table position and moisture can constitute an indicator of fire risk, understanding the water balance dynamics of peatland dominated basins may help in managing fire risk (Kartiwa et al., 2023; Mortelmans et al., 2024). In October 2022, the Scotty Creek basin was impacted by a late-season wildfire. While the wetland flux tower and several cutthroat flumes remained intact (Figure S1), the landscape flux tower was destroyed and rebuilt in March 2023. Our work, which contributes to understanding the hydrological response of a rapidly thawing boreal peatland complex, can serve as a baseline for understanding the combined effects of permafrost thaw accelerated by wildfire."

**Finally, there is very limited discussion of error or uncertainty here. I am not advocating for a full uncertainty or error analysis, but the authors could be a bit more forthcoming about this, particularly given the levels of interpolation and reliance on point measurements. Again, I understand the authors know the data well, but I'm somewhat surprised that there is little discussion on this considering some things like SWE come from a site far from the basins. I'm sure there has been some assessment done as to how effective this is other than citing another paper.**

We have modified the Methods section to better emphasize that all data presented at sub-basin scale including snow water equivalent (SWE) were measured within or near the three sub-basins in both the wetlands and the forest:

Starting at line 301 (original submission) and 312 (revised submission): "Precipitation ($P_{SUB\text{-}BASIN}$) including R and SWE in late March just before the start of snowmelt (i.e., $SWE_{MAX}$) was obtained from rain gauge measurements ($R_{WEST} = R_{EAST} = R_{SOUTH}$), and calculated as weighted mean for each sub-basins ($SWE_{MAX\_SUB\text{-}BASIN}$) according to sub-basin specific cover areas (i.e., wetland [$A_{WET}$] and forest areal coverage [$A_{FOR}$]) and associated measured SWE from late-winter snow surveys (i.e., forest [$SWE_{MAX\_FOR}$] and wetland SWE [$SWE_{MAX\_WET}$]), respectively:"

To better constrain the sub-basin water balance in May, the snowmelt rate for the 2014-2016 period was estimated using a simple snowmelt model based on the temperature index equation (see the following response).

At the basin scale, we chose to use SWE data from Fort Simpson due to time series continuity and their strong correlation with SWE data from Scotty Creek (Connon et al., 2021). Fort Simpson A station is operated by NAVCAN for the purposes of air traffic safety, while the

'Climate' station is operated by ECCC for the purposes of climate monitoring. Although the two stations are located right beside each other, the gauges are different, and the recorded values are different. The Climate station likely produces more reliable data. We slightly refined the Methods section by adding information on data handling:

Starting at line 133 (original submission) and 137 (revised submission): "Data from the Fort Simpson Climate station, WMO ID: 71365, was gap-filled with data from the Fort Simpson A station, WMO ID: 71946, Environment and Climate Change Canada, climate.weather.gc.ca."

**Could you not have used a snowmelt model? You use a complex model for ET at the basin scale so I'm unsure why you can't apply a simple melt model to appropriately partition SWE at least at the monthly level.**

At the basin scale, since we reported the annual (hydrological year) water balance, snowmelt timing was not an issue. At the sub-basin scale, accounting for all the SWE in May resulted in an overestimation of its contribution for that month, however, this was discussed in this study. Incorporating a snowmelt model would reduce the water balance residual in May but would not provide additional insights within the scope of this study. We performed a simple snowmelt model using the temperature index equation (Pomeroy and Brun, 2001; Fontrodona-Bach et al., 2025):

Snowmelt (mm of water equivalent) = $C_f \times (T_{air} - T_{threshold})$

where, $C_f$ is the melt factor (mm °C$^{-1}$ day$^{-1}$), $T_{air}$ is the daily mean air temperature measured at Scotty Creek, $T_{threshold}$ is the threshold temperature at which snow begins to melt. $T_{threshold}$ ranged from -1 to +1 °C (Fontrodona-Bach et al., 2025 [preprint], https://doi.org/10.5194/egusphere-2025-1214 and references therein).

$C_f$ was estimated using the relationship between snow density and $C_f$ described by Rango and Martinec (1995) (10.1111/j.1752-1688.1995.tb03392.x). Given that snow density at Scotty Creek ranges from 0.11 to 0.29 (Connon et al,. 2021), the corresponding $C_f$ values were estimated to range from 1 to 3 mm °C$^{-1}$ day$^{-1}$.

To account for these uncertainties, we performed 10,000 simulations using a Monte Carlo approach, where $C_f$ and $T_{thresholds}$ were randomly sampled within their respective ranges (Figure 5, added to the Supplementary Material as Figure S3).

[Figure]

Figure 5. Cumulative snowmelt evolution estimates for 2014, 2015, and 2016. The colored line represents the median of the 10,000 simulations, while the shaded area represents the interquartile range (25th to 75th percentiles). The vertical black dashed line marks May 1st.

Accordingly, we added Figure 5 and the snowmelt model description to the Supplementary Material and added the snowmelt model outputs in the discussion as follows (section 4.1):

Starting at line 514 (original submission) and 522 (revised submission): "Regarding monthly water balance, high residuals observed in May for all three years (Figure 4-a, b, c) might be explained by the inclusion of snowmelt input through SWE that month. Due to limited data availability, $SWE_{MAX}$, estimated in late March just before the onset of snowmelt, served as a proxy for snowmelt input in the water balance in May, highlighting the challenge of appropriately accounting for the spring freshet in the growing season water balance through observations. To shed light on this challenge, we estimated the amount of snowmelt at the end of April using a simple temperature index model (Figure S3). The estimated snowmelt amounts (median [25th-75th percentiles] from 10,000 Monte Carlo simulations) at the end of April were 105 [78-136] mm in 2014, 187 [138-238] mm in 2015, and 125 [92-159] mm in 2016. These ranges correspond closely to the $SWE_{MAX}$ measured each year (102 mm, 167 mm and 128 mm in 2014, 2015 and 2016, respectively), suggesting that only a small portion of $SWE_{MAX}$ contributed to the May water balance. This would reduce the high residuals in the May water balance in the West sub-basin estimated as +149 mm (2014), +176 mm (2015) and +117 mm (2016)."

**Line 503-504. This value for black spruce transpiration seems unusually low and does not agree with more modern and methodologically detailed work of Perron et al. (2023) from Scotty Creek.**

We agree with the reviewer.

Starting at line 502 (original submission) and 509 (revised submission): "For example, higher wetland (2.9 ± 1 mm day$^{-1}$) than forest ET (1.7 ± 0.6 mm day$^{-1}$) at Scotty Creek was reported for June-mid July 2013 (Warren et al., 2018), with transpiration from black spruce and tamarack accounting for only approximately 6 % to 12 % of forest ET (Perron et al., 2023)."

**Do you examine any patterns in the timing of the hydrograph at the basin scale? You suggest that snowmelt has advanced 25 days since an earlier period? Perhaps no but if you discuss it I'm unsure why you didn't evaluate it.**

We reported a 25-day advance in snowmelt based on the second half of the 21st century in an Arctic tundra landscape (Pohl et al., 2007). At Scotty Creek, Chasmer et al. (2017) observed an earlier snow melt (16 days) during the 2000-2009 period compared to 1970-1979. We concurrently observed that at Scotty Creek runoff increased approximately 15 days earlier during the 2009-2022 period compared 1995-2008 (Figure 6, added to the Supplementary Material as Figure S4). However, this earlier increase is modest, i.e. at only ten liters per second.

[Figure]

Figure 6. Composite hydrographs for 14-year periods of 1995-2008 and 2009-2022 at Scotty Creek basin outlet. Colored bands correspond to 95 % confidence intervals. The inlet shows a zoom on the spring freshet onset.

Additionally, we observe that the spring freshet peak at the Scotty Creek basin outlet has increased from 3.7 to 4.5 m$^3$ s$^{-1}$ over the 1995-2008 to 2009-2022 periods, respectively (Figure 6). Similar observations were made at the Jean Marie River basin outlet, close to Scotty Creek (Connon et al., 2021).

As snow melts more rapidly in wetlands than in forests, the high rate of wetland expansion associated with forest loss might favor an earlier and shorter snowmelt in the future (Quinton et al., 2019). Consequently, we modified as follows in the manuscript:

Starting at line 518 (original submission) and 533 (revised submission): "Despite the observational challenges, particular attention should be paid to this snowmelt period, which is profoundly influenced by climate warming. Firstly, the spring freshet is shown to occur earlier in the Arctic-boreal region (Chasmer and Hopkinson, 2017; Mack et al., 2021; Pohl et al., 2007; Woo et al., 2008). At Scotty Creek, an earlier snowmelt of 16 days was observed during the 2000-2009 period compared to the 1970-1979 period (Chasmer et al., 2017). Consistently, the Scotty Creek basin hydrograph analysis revealed an earlier increase in discharge (~15 days) during the 2009-2022 period compared to the 1995-2008 period (Figure S4). Secondly, earlier snowmelt leads to a longer snowmelt period, as projected for the Liard River watershed, resulting in a more gradual snowmelt (Woo et al., 2008). However, an increase in wetland extent caused by forested peat plateau collapse can contribute to shorter snowmelt period since snow melts faster in wetlands than in forest stands (Connon et al., 2021; Quinton et al., 2019). Shorter snowmelt periods can result in higher spring freshet peaks, as observed at Scotty Creek and the adjacent Jean Marie River meso-scale basin (Connon et al., 2021)."

**Figure 2 - it is a bit hard to see some of the runoff values or distinguish them, particularly in 2022. Some of this is due to the distracting nature that the uncertainty in the runoff area for EAST - perhaps think of a way to make this more clear? Two lines? A dashed line? I'll leave it up to the authors but the shaded yellow is a bit distracting.**

We replaced the yellow band for the East sub-basin by two lines representing the runoff from two drainage area estimates, i.e., from this study (yellow line) and Connon et al. (2015) (dashed line). We lightened the shade of green for ET and darkened the shade of gray for basin-scale runoff to improve clarity and contrast. Below the updated figure:

[Figure]

**Why was the BESS data not corrected to the observations? Claiming that landcover differences are the reason that it underestimates ET is simply guessing and can easily be tested. My guess is that like many land models it is just underestimating ET.**

We did not correct the BESS data to the observations because the monthly relationship between measured ET and BESS ET was based on only three growing seasons (May–September, 2014–2016), with a relatively low coefficient of determination ($R^2 = 0.44$; see Figure 5-b in the manuscript).

Additionally, independent estimates by Hayashi et al. (2004), based on a chemical method at the basin scale, reported annual ET values between 280 and 300 mm for 1999–2002. These

are consistent with BESS estimates over our study period (223–311 mm; mean ± std: 261 ± 22 mm), supporting the overall magnitude of the BESS ET estimates.

While land cover differences—such as more mineral uplands and fewer wetlands in the northern part of the Scotty Creek basin—may contribute to lower BESS ET, underestimation was also observed in the headwater sub-basins. This suggests that the bias may be at least partly inherent to the BESS model rather than solely due to land cover differences. However, the pixels representing the headwater portion cover a larger area than the three sub-basins.

We refined the BESS ET estimate for the headwater portion by updating the number of pixels considered. One pixel located outside the basin boundary was removed (see Figure S2 of the Supplementary Material). This correction did not affect the ET values.

We have modified the corresponding paragraph in Section 4.2 of the manuscript as follows:

Starting at line 554 (original submission) and 577 (revised submission): "Our results show that modeled ET obtained with the BESS model at basin scale underestimated (annually ~100 mm) observed ET (Figure 5-b). Given that wetland ET is higher than forest ET (Helbig et al., 2016b; Perron et al., 2023), the underestimation of ET might be related to land cover heterogeneity at basin scale. The northern, i.e., downstream, portion of the basin is dominated by mineral uplands with better drainage and mainly covered by deciduous or mixed forest stands (Chasmer et al., 2014). Consistently, ET estimations from a chemical method at the Scotty Creek basin scale ranged from 280 to 300 mm year$^{-1}$ for the 1999-2002 period (Hayashi et al., 2004). However, modeled ET was lower than observed ET at the sub-basin scale, probably underestimating the contribution of wetlands (Figure 5-c). Although this difference may stem from tendency of the BESS model to underestimate the spatial variability of ET in wetland-rich landscapes such as boreal peatland complexes near the southern permafrost limit, we cannot disentangle the extent to which it reflects a general underestimation of ET versus a specific underestimation in wetlands."

**Check colour scheme on Figure 6 (Rain vs SWE and 2022 which doesn't match the others)**

We modified the legend to better match the figure:

[Figure]

**REVIEWER 2**

Lhosmot et al., present water balance studies for a well-instrumented and studied permafrost-affected boreal peatland complex. They quantify the impact of uncertainty in catchment delineation on water budgets, and compare water budgets carried out at different scales and with locally-observed and publicly available observational and simulated data. The find, among other things, that evapotranspiration is the dominant outgoing water flux but that this variable is considerably underestimated in publicly available simulated data. The results also highlight the difficulty of obtaining exact estimates of water fluxes such as discharge in a low relief landscape which is hydrologically dominated by spring freshet, as illustrated by high water balance residuals during this period.

The study provides new and valuable insights on the hydrologic behaviour of rapidly changing permafrost peatland complexes, as well as on the limitations of what information can be retrieved from detailed field-observations of water balance components as well as publicly available monitoring data. I find that the scientific rigour is high, based on the choice and descriptions of methods and the tight link to previous relevant studies from the same, and other similar areas. I have a few minor comments that I recommend that the authors address before publication of this nice manuscript.

Thanks.

**L190: It is unclear to me if this instrument failure regards just one or both of the eddy covariance stations. Recommend clarification.**

We clarified as follows:

Starting at line 188 (original submission) and 192 (revised submission): "Due to instrument failure on the landscape flux tower, $CO_2$ and $H_2O_{(g)}$ molar densities were measured with an enclosed $CO_2/H_2O_{(g)}$ infrared gas analyzer (LI7200; LI-COR Biosciences Inc., Lincoln, NE) between March and August 2015."

**Section 2.4: I read it this section several times to understand what was done here, so there might be a good idea to see if it is possible to make this easier to read (although, I might just have been tired!). On L246, it is stated that 15 rating curves were obtained, one for each flume 2014-2016. There were five flumes, so I suppose that you mean one rating curve per flume and year 2014-2016?**

We corrected the manuscript text as follows:

Starting at line 246 (original submission) and 249 (revised submission): "Twelve rating curves to convert WTP to half-hour discharge estimates were obtained from manual discharge and WTP measurements made during and shortly after snowmelt in late April to early May (spring freshet) and late May (baseflow) in 2014-2016, respectively. For the West sub-basin, we used one rating curve per year for each of the two outlets (West1 and West2), thus six rating curves in total. For the South sub-basin, we used one rating curve per year at the South1 outlet (thus three rating curves in total) and a single rating curve at the South2 outlet, created in 2015 and used for all three years. The East sub-basin consisted of one outlet, which was monitored in 2014 and 2015, with one rating curve per year (no data was available for 2016)."

**I'm also curious about the gap-filling, especially since the West sub-basin series had 75% of data gap-filled by method 1 plus 14% gap-filled by method 2 (if I understand the text correctly). This adds up to almost 90% gap-filled data for this sub-basin, yet this is the sub-basin that you choose to show monthly water balance for in section 3.3. A motivation for your trust in this gap-filled data is warranted, as is a motivation for choosing to focus on this particular sub-basin in section 3.3, given this gap-filled data. Can you show that the rating curves from wetland WTP generate data that is well correlated with that from original rating curves?**

Gap-filling percentages were first re-examined, and an error resulting in minor inaccuracies in the reporting of gap-filled flow rate data across all sub-basins was corrected. The updated values are now reported in the manuscript accordingly (see Table below). However, the issue that reviewer 2 has rightly underlined remains valid: the west sub-basin is still 79 % gap-filled.

Here are the updated values of discharge gap-filling percentage:

| Sub-basin | no Gap-filling | Gap-filling method 1 | Gap-filling method 2 |
|---|---|---|---|
| West | 21 | 79 | 0 |
| East | 88 | 3 | 9 |
| South | 67 | 14 | 19 |

We chose the West sub-basin as this study's primary focus because it is situated within the footprints of the landscape and wetland flux towers. Most West sub-basin gap-filled values occurred during base flow, but 41 % of the time series gap-filled during the 2016 freshet. Method 1 gap-filling was performed using a piecewise monthly linear relationship between wetland WTP and outlet WTP (from May to September each year), with an average $R^2$ of 0.70 ± 0.33 (std). We have modified the manuscript accordingly (starting line 260 of the revised manuscript).

**L269: I'm curious why the average 2002-2022 ET was used to calculate the 1996-2001 water balance? If ET data for 1996-2001 was not available, how much can you still say about the basin water balance for those years, considering that ET is the dominant flux in the basin? This is relevant for results presented on L477, if change in storage at the basin level is calculated as the residual of water balance fluxes.**

MODIS data, used as atmosphere and land input data for the BESS model, did not exist before 2001. We used the average 2002-2022 to estimate 1996-2001 ET because the interannual variability of ET was low according to BESS values from 2002 to 2022 (ranges between 223 to 311 mm [mean ± std, 261 ± 22 mm]). Estimating ET from 1996 to 2001 allows us to at least approximate variations in water storage for the 1996-2001 period. Accordingly, in Figure 6 of the manuscript, the estimated variations in ET and water storage for 1996-2001 are represented by a dashed line.

**Table 2: The signs for ET and Q are inconsistent for East sub-basin, relative to other sub-basins.**

This issue has been addressed, and Table 2 has been moved to the Supplementary Material as Table S1, in accordance with the reviewer's comments.

**L508: typo, at end of line "the could"?**

Corrected.

**L543: Suggestion, if you find relevant in context of your results. Another potential mechanism was presented by (Jutebring Sterte et al., 2018, 2021), for a boreal catchment in Sweden. They showed that seasonal freezing of the wetland resulted in high wetland runoff during freshet, while soil frost in the (less saturated) forest soils had less impact on drainage dynamics.**

Thank you for these references. These results can definitely contribute to explaining the difference of freshet peak between the basin and the sub-basins in 2014. Corrected:

Starting at line 539 (original submission) and 557 (revised submission): "However, an exception occurred during the driest year (2014) when the peak in basin runoff peak was more than ten times lower than for the sub-basins (Figure 2). This difference might be partially explained by the higher proportional coverage of wetlands in the headwater sub-basins (~40 %) compared to the entire basin (~20 %) and high coverage of mineral uplands in the basin (~40 %; Chasmer et al., 2014). More water is expected to be stored in saturated wetlands than in mineral uplands (McCarter et al., 2020; Price, 1987), which may help sustain a higher runoff ratio during years with low late-winter SWE, as observed in 2014. The higher degree of saturation in wetlands compared to mineral uplands can favour surface

runoff over water infiltration during the spring freshet, as observed in small-scale basins in Sweden (Jutebring Sterte et al., 2018, 2021). Dry conditions in 2013 (annual total P = 259 mm, Figure 6) may have further exacerbated the drying of mineral uplands compared to wetlands, thereby enhancing infiltration at the basin scale during the 2014 snowmelt."

---

## Author Response (AR2)

Dear Associate Editor,

We are grateful for the supportive and constructive comments provided by you and the two reviewers on our manuscript. We have addressed the remarks from Reviewer 1 and incorporated the suggested changes into the manuscript.

Below, you will find the original comments from Reviewer 1 in bold, followed by our detailed responses and a description of the revisions made to the manuscript.

Kind regards,

Alexandre Lhosmot, on behalf of all authors

**The authors have made substantial efforts to improve the manuscripts based on the referees' comments on the previous version of the manuscript. I have only a few minor comments on this revised manuscript:**

**L262: a standard deviation (std = 0.33) is given after the R2 value (0.7) of the regression of outlet and wetland WTP. I'm not sure what this std refers to, as R2 values are not associated with this statistic.**

Thank you for your comment. To clarify, the $R^2$ value was computed separately for each month from May to September over the three years of the study, but only for the months when both outlet and wetland WTP data were available, resulting in a total of 8 $R^2$ values. The mean (0.70) and standard deviation (0.33) refer to the distribution of these 8 $R^2$ values. We agree that standard deviation is not associated with a single $R^2$, but here it describes the variability of $R^2$ across time. We propose to clarify this sentence as follows:

Starting at line 261 (revised submission): "The mean coefficient of determination, $R^2$, (± std) of the monthly linear relationships between wetland and outlet WTP, calculated for the months with available data between May and September over the three study years, was 0.70 ± 0.33 (n = 8)."

**L360: "The annual total P in 2016 and 2015 was lower or higher than the 27-year mean". Lower or higher? Which is it?**

We corrected as follows:

Starting at line 261 (revised submission): "The annual total P in 2016 and 2015 was lower and higher than the 27-year mean, respectively, but within one std."

**L534: "Firstly, the spring freshet is shown to occur earlier in the Arctic-boreal region." Earlier than what? I assume earlier than before, but this is not stated.**

Thanks, we corrected as follows:

Starting at line 535 (revised submission): "Firstly, the spring freshet in recent years is shown to occur earlier in the Arctic-boreal region compared to previous decades."